

# Real-time dynamics of the $O(4)$ scalar theory within the fRG approach

Yang-Yang Tan[1], Yong-Rui Chen[1] and Wei-Jie Fu[1,2]⋆

**1** School of Physics, Dalian University of Technology, Dalian, 116024, P.R. China
**2** Institute of Theoretical Physics, Chinese Academy of Sciences, Beijing, 100190, P.R. China

⋆ wjfu@dlut.edu.cn

## Abstract

In this paper, the real-time dynamics of the $O(4)$ scalar theory is studied within the functional renormalization group formulated on the Schwinger-Keldysh closed time path. The flow equations for the effective action and its $n$-point correlation functions are derived in terms of the "classical" and "quantum" fields, and a concise diagrammatic representation is presented. An analytic expression for the flow of the four-point vertex is obtained. Spectral functions with different values of temperature and momentum are obtained. Moreover, we calculate the dynamical critical exponent for the phase transition near the critical temperature in the $O(4)$ scalar theory in $3 + 1$ dimensions, and the value is found to be $z \simeq 2.023$.



# 1 Introduction

The past years have seen rapid progress in our understanding of the strongly correlated physics and its in-medium effects in the context of Euclidean field theories at finite temperature and density, e.g., QCD on a discretized lattice of Euclidean space and time [1, 2], functional continuum QCD within the functional renormalization group (fRG) [3–5] and Dyson-Schwinger equations (DSE) [6–10]. Relevant studies have provided us with a plethora of properties of QCD at finite temperature and density, such as equation of state, thermodynamics, fluctuations, phase structure and so forth. Exploration of other properties of the same importance, for instance, nonequilibrium time evolution of quantum fields far away from the thermal equilibrium [11], dynamics of critical fluctuations [12], spectral functions and transport coefficients [13], dynamic critical exponents [14], etc., is, however, beyond the capability of the Euclidean field theories, and direct computations of field theories in the Minkowski spacetime are indispensable.

The formalism of the functional integral on a closed time path [15, 16], i.e., the Schwinger-Keldysh path integral, is well suited for investigations of the above-mentioned properties of real-time dynamics, and also see, e.g., [11, 13, 17, 18] for relevant reviews. It has proved to be a powerful tool to deal with both the equilibrium and nonequilibrium thermodynamic systems. Unfortunately, lattice Monte-Carlo simulations in the formalism of Keldysh path integral are hindered by the notorious 'sign' problem, and therefore, in order to study observables related to nonperturbative real-time dynamics, e.g., spectral functions in QCD or other strongly correlated system [19, 20], one has to resort to functional continuum methods. In the references above, a spectral DSE approach in terms of Källén-Lehmann representation of correlation functions is put forward, and is applied in the computation of spectral functions in the $\phi^4$-theory and the ghost spectral function in Yang-Mills (YM) theory.

The functional renormalization group is a nonperturbative approach of continuum field theories. In fRG, quantum fluctuations of different momentum shells are integrated out successively via running of flow equations, and thus it is very convenient to cope with physical problems involving different degrees of freedom on different scales [21], see also e.g., [22–29] for QCD related reviews. Remarkably, significant progress have been made over the last several years in the first-principle fRG computation of QCD or YM theory in the vacuum [30–34] and at finite temperature and density [3, 5, 35], in the formalism of Euclidean path integral. In the meanwhile, relevant studies in the low energy effective field theories also provided us with a wealth of useful information on QCD phase structure [36–39] equation of state [40, 41], baryon number fluctuations [42–53], baryon-strangeness correlations [54–56], critical exponents [57], etc.

A promising and intriguing possibility is to combine fRG with the Keldysh path integral, i.e., formulating flow equations in terms of the functional integral of closed time path. One of the relevant pioneer works has been done in [58], where the fRG on a closed time path is employed to study nonthermal fixed points of the $O(N)$ scalar theory, see also [59]. Another conceptually different combination between fRG and the Keldysh path integral is put forward in [60–62], where the regulation is implemented on the time rather than the renormalization group (RG) scale, such that a time evolution equation for the non-equilibrium effective action is obtained. Furthermore, the transition from unitary to dissipative dynamics is investigated in the framework of the real-time fRG [63]. Very recently, spectral functions for the scalar field theory in $d=0+1$ dimensions are calculated within the fRG formulated on the Keldysh path [64]. The fRG with the Keldysh functional integral has also been used in open quantum systems to study e.g., nonequilibrium transport [65], dynamical critical behavior [66], etc., and see e.g., [18, 29] for more comprehensive discussions.

In this work we would like to adopt the fRG formulated on the Keldysh path, to investigate

the real-time dynamics of the $O(4)$ scalar theory. We will calculate the spectral functions in thermal equilibrium. Note that calculations of the spectral functions in thermal equilibrium have attracted lots of attentions in recent years. While ill-defined, construction of the spectral functions from Euclidean data sheds new light on time-like properties of correlation functions [67,68]. Within some specific truncations in fRG in imaginary time, it is possible to analytically continue the Euclidean flow equation into the Minkowski one on the level of analytic equations, see, e.g., [69–72] for more details. Furthermore, we will also investigate the dynamical critical exponent near the phase transition in the $O(4)$ scalar theory in the formalism of real-time fRG.

This paper is organized as follows: In Section 2 we give a brief introduction about the formalism of the fRG with the Keldysh functional integral in the context of the $O(N)$ scalar theory, including notations and Feynman rules. The flow of effective potential is discussed in Section 3. In Section 4 we give the flow equations for the propagators and vertices, and describe the relevant truncations. In Section 5 we present and discuss our numerical results. A summary with conclusions is given in Section 6. Technical details regarding the flow equations are presented in the appendices. In Appendix A we give a derivation of the fRG flow within the Keldysh functional integral. The explicit formulae for function $I_k(p)$ in Equation (60) are collected in Appendix B.

## 2 The $O(N)$ scalar theory within the real-time fRG approach

In this section we begin with a RG scale $k$-dependent effective action for the real-time $O(N)$ scalar theory as follows

$$\Gamma_k[\phi_c, \phi_q] = \int_x \left[ Z_{\phi,k}(\partial_\mu \phi_q) \cdot (\partial^\mu \phi_c) - U_k(\phi_c, \phi_q) \right], \tag{1}$$

where $\phi_{i,c}$ and $\phi_{i,q}$ ($i = 0, 1, ... N-1$) are the "classical" and "quantum" scalar fields with $N$ components, respectively. In Equation (1) we have adopted a local potential approximation (LPA) with a $k$-dependent wave function renormalization $Z_{\phi,k}$, and it allows us to introduce the fRG formalism of Keldysh fields conveniently, which can also be easily extended to cases beyond LPA. The effective potential in Equation (1) is given by

$$U_k(\phi_c, \phi_q) = V_k(\rho_+) - V_k(\rho_-), \tag{2}$$

with $\rho_\pm = \phi_\pm^2/2$, which is obviously $O(N)$ invariant. Here, $\phi_\pm$ denote the fields on the forward and backward branches in the formalism of Keldysh field theory, cf. Appendix A for details.

On the equations of motion (EoM) of fields, i.e., the expectation values of fields under vanishing external sources as shown in Equation (94), that would be denoted by variables with a bar in what follows, the "quantum" field is obviously vanishing, viz.,

$$\bar{\phi}_q|_{\text{EoM}} = 0. \tag{3}$$

Furthermore, we would like to investigate the breaking of $O(N)$ symmetry into the $O(N-1)$ one. To that end, a nonvanishing value for one component of the "classical" field is introduced as follows

$$\bar{\phi}_c|_{\text{EoM}} = \begin{cases} \bar{\phi}_{0,c} & i = 0 \\ 0 & i \neq 0 \end{cases}, \tag{4}$$

where the direction of zero component is chosen to that of the symmetry breaking. Consequently, one can define the sigma and pion fields as follows

$$\sigma_c = \phi_{0,c} - \bar{\phi}_{0,c}, \qquad \sigma_q = \phi_{0,q}, \tag{5}$$

$$iG^R_{\sigma,k} = \text{------} \underset{c}{\bullet}\,\,\,\underset{q}{} \,, \quad iG^A_{\sigma,k} = \text{------} \underset{q}{}\,\,\,\underset{c}{} \,, \quad iG^K_{\sigma,k} = \text{----}\underset{c}{}\,\underset{q\,\,q}{\circ}\,\underset{c}{}\text{----} \,,$$

$$i(G^R_{\pi,k})_{ij} = \overset{i}{\text{------}}\underset{c}{}\,\,\,\underset{q}{\overset{j}{}} \,, \quad i(G^A_{\pi,k})_{ij} = \overset{i}{\text{------}}\underset{c}{}\,\,\,\underset{q}{\overset{j}{}} \,, \quad i(G^K_{\pi,k})_{ij} = \overset{i}{\text{----}}\underset{c}{}\,\underset{q\,\,q}{\circ}\,\underset{c}{\overset{j}{}}\text{----}$$

Figure 1: Diagrammatic representation of the retarded, advanced, and Keldysh propagators for the $\sigma$ and $\pi$ mesons. The retarded and advanced propagators are denoted by a dashed line with two points labelled with "$c,q$" and "$q,c$", respectively. The Keldysh propagator is represented by a line with an empty circle inserted in-between.

and

$$\pi_{i,c} = \phi_{i,c} \,, \qquad \pi_{i,q} = \phi_{i,q} \,, \tag{6}$$

with $i = 1, ... N - 1$. Note that the nonvanishing $\bar{\phi}_{0,c}$ has been shifted away in the definition of the "classical" sigma field in Equation (5).

The effective action in Equation (1) is straightforwardly reformulated in terms of the newly defined $\sigma$ and $\pi$ fields, which reads

$$\Gamma_k[\phi_c, \phi_q] = \int_x \left\{ \left[ Z_{\phi,k}\partial_\mu \sigma_q \partial^\mu \sigma_c - m^2_{\sigma,k}\sigma_q \sigma_c \right] + \left[ Z_{\phi,k}(\partial_\mu \pi_q)\cdot(\partial^\mu \pi_c) - m^2_{\pi,k}\pi_q \cdot \pi_c \right] \right.$$
$$- \frac{\lambda_{3\sigma,k}}{6}(3\sigma_q \sigma_c^2 + \sigma_q^3) - \frac{\lambda_{1\sigma 2\pi,k}}{2}\left[ 2\sigma_c(\pi_q \cdot \pi_c) + \sigma_q(\pi_q^2 + \pi_c^2) \right]$$
$$- \frac{\lambda_{4\sigma,k}}{6}\sigma_q \sigma_c(\sigma_q^2 + \sigma_c^2) - \frac{\lambda_{2\sigma 2\pi,k}}{2}\left[ \sigma_q \sigma_c(\pi_q^2 + \pi_c^2) + (\pi_q \cdot \pi_c)(\sigma_c^2 + \sigma_q^2) \right]$$
$$\left. - \frac{\lambda_{4\pi,k}}{6}(\pi_q \cdot \pi_c)(\pi_q^2 + \pi_c^2) - \lambda_{1\sigma,k}\sigma_q + \cdots \right\} \,, \tag{7}$$

where the effective potential in Equation (1) has been expanded up to the fourth order in powers of fields, and the $\sigma$ and $\pi$ masses are given by

$$m^2_{\sigma,k} \equiv V'_k(\bar{\rho}_c) + 2\bar{\rho}_c V^{(2)}_k(\bar{\rho}_c) \,, \tag{8}$$

$$m^2_{\pi,k} \equiv V'_k(\bar{\rho}_c) \,, \tag{9}$$

with $\bar{\rho}_c \equiv \bar{\phi}_c^2/4$. The three-meson couplings in Equation (7) read

$$\lambda_{3\sigma,k} \equiv 3\bar{\rho}_c^{\frac{1}{2}}V^{(2)}_k(\bar{\rho}_c) + 2\bar{\rho}_c^{\frac{3}{2}}V^{(3)}_k(\bar{\rho}_c) \,, \tag{10}$$

$$\lambda_{1\sigma 2\pi,k} \equiv \bar{\rho}_c^{\frac{1}{2}}V^{(2)}_k(\bar{\rho}_c) \,, \tag{11}$$

and the four-meson couplings

$$\lambda_{4\sigma,k} \equiv \frac{3}{2}V^{(2)}_k(\bar{\rho}_c) + 6\bar{\rho}_c V^{(3)}_k(\bar{\rho}_c) + 2\bar{\rho}_c^2 V^{(4)}_k(\bar{\rho}_c) \,, \tag{12}$$

$$\lambda_{4\pi,k} \equiv \frac{3}{2}V^{(2)}_k(\bar{\rho}_c) \,, \tag{13}$$

$$\lambda_{2\sigma 2\pi,k} \equiv \frac{1}{2}V^{(2)}_k(\bar{\rho}_c) + \bar{\rho}_c V^{(3)}_k(\bar{\rho}_c) \,. \tag{14}$$

Note that in the last line of Equation (7) there is a term linear in $\sigma_q$ with the relevant coefficient given by

$$\lambda_{1\sigma,k} \equiv 2\bar{\rho}_c^{\frac{1}{2}} V_k'(\bar{\rho}_c). \tag{15}$$

The infrared (IR) regulator term as shown in Equation (100) in our case reads

$$\Delta S_k[\phi_c, \phi_q] = \int \frac{d^4q}{(2\pi)^4} \Big[ \sigma_q(-q) R_{\sigma,k}(q) \sigma_c(q) + \pi_{i,q}(-q) \big(R_{\pi,k}\big)_{ij}(q) \pi_{j,c}(q) \Big], \tag{16}$$

where we have used a 3$d$ flat regulator [73, 74] in this work, to wit,

$$R_{\sigma,k}(q) = R_{\phi,k}(q) = Z_{\phi,k}\Big( -\boldsymbol{q}^2 r_B\big(\frac{\boldsymbol{q}^2}{k^2}\big) \Big), \tag{17}$$

and

$$\big(R_{\pi,k}\big)_{ij}(q) = R_{\phi,k}(q)\delta_{ij}, \tag{18}$$

with

$$r_B(x) = \Big(\frac{1}{x} - 1\Big)\Theta(1-x), \tag{19}$$

and here $\Theta(x)$ is the Heaviside step function. Thus, the regulator matrix, cf. Equation (101), in the bases of fields $\Phi = (\sigma_c, \{\pi_{i,c}\}, \sigma_q, \{\pi_{i,q}\})$ with $i = 1, ... N-1$ is readily obtained as

$$R_k(q) = \begin{pmatrix} 0 & R_k^A(q) \\ R_k^R(q) & 0 \end{pmatrix}, \tag{20}$$

with

$$R_k^R(q) = R_k^A(q) = \begin{pmatrix} R_{\sigma,k}(q) & 0 \\ 0 & R_{\pi,k}(q) \end{pmatrix} \equiv \hat{R}_{\phi,k}(q). \tag{21}$$

Note that in Equation (20) we do not include any regulator for the $qq$-component, since only the real parts of two-point functions are regulated in this work. This is adequate for cases in thermal equilibrium, where the Keldysh propagator is related to the retarded and advanced propagators by the fluctuation-dissipation relation as shown in Equation (36) in the following. But if the regulator in Equation (17) is extended to the one having a finite imaginary part, as done in some nonequilibrium calculations, e.g. [75], a nonvanishing $qq$-component of regulators is necessary.

With the regulator in Equation (21), one can reformulate the flow equation for the effective action in Equation (113) as such

$$\partial_\tau \Gamma_k[\Phi] = \frac{i}{2} \text{STr}\Big[ \tilde{\partial}_\tau \ln\big( \Gamma_k^{(2)}[\Phi] + R_k \big) \Big], \tag{22}$$

where $\tilde{\partial}_\tau$ indicates that $\partial_\tau$ hits only the regulator in Equation (20), and $\tau \equiv \ln(k/\Lambda)$ is the RG time, with an initial evolution scale $\Lambda$, i.e., the ultraviolet (UV) cutoff. In Equation (22) one has employed the notation as follows

$$\big(\Gamma_k^{(2)}[\Phi]\big)_{ab} \equiv \frac{\delta^2 \Gamma_k[\Phi]}{\delta \Phi_a \delta \Phi_b}. \tag{23}$$

Moreover, it is more convenient to make the reorganization as follows

$$\Gamma_k^{(2)} + R_k = \mathcal{P}_k + \mathcal{F}_k, \tag{24}$$

where $\mathcal{P}_k$ is the matrix of inverse propagators with regulators, and $\mathcal{F}_k$ is the interaction sector which encodes the field dependence.

In what follows we consider the $O(N)$ scalar theory in thermal equilibrium with a temperature $T$. As a consequence, one arrives at

$$\mathcal{P}_k = \begin{pmatrix} 0 & \mathcal{P}_k^A \\ \mathcal{P}_k^R & \mathcal{P}_k^K \end{pmatrix}. \tag{25}$$

Here the inverse retarded propagator reads

$$\mathcal{P}_k^R = \begin{pmatrix} \mathcal{P}_{\sigma,k}^R & 0 \\ 0 & \mathcal{P}_{\pi,k}^R \end{pmatrix}, \tag{26}$$

with

$$\mathcal{P}_{\sigma,k}^R = Z_{\phi,k}\left[ q_0^2 - \boldsymbol{q}^2\left(1 + r_B\left(\frac{\boldsymbol{q}^2}{k^2}\right)\right)\right] - m_{\sigma,k}^2 + \mathrm{sgn}(q_0)i\epsilon, \tag{27}$$

$$\left(\mathcal{P}_{\pi,k}^R\right)_{ij} = \left\{ Z_{\phi,k}\left[ q_0^2 - \boldsymbol{q}^2\left(1 + r_B\left(\frac{\boldsymbol{q}^2}{k^2}\right)\right)\right] - m_{\pi,k}^2 + \mathrm{sgn}(q_0)i\epsilon \right\}\delta_{ij}, \tag{28}$$

where the infinitesimal terms with a sign function are used to determine the contour for the retarded propagator in the complex plane of $q_0$. The advanced counterpart $\mathcal{P}_k^A$ in Equation (25) is related to $\mathcal{P}_k^R$ through a complex conjugate, i.e.,

$$\mathcal{P}_k^A = (\mathcal{P}_k^R)^*. \tag{29}$$

The Keldysh component of the inverse propagator in Equation (25) is given by

$$\mathcal{P}_k^K = \begin{pmatrix} \mathcal{P}_{\sigma,k}^K & 0 \\ 0 & \mathcal{P}_{\pi,k}^K \end{pmatrix}, \tag{30}$$

with

$$\mathcal{P}_{\sigma,k}^K = 2i\epsilon\,\mathrm{sgn}(q_0)\coth\left(\frac{q_0}{2T}\right), \tag{31}$$

$$\left(\mathcal{P}_{\pi,k}^K\right)_{ij} = \left[2i\epsilon\,\mathrm{sgn}(q_0)\coth\left(\frac{q_0}{2T}\right)\right]\delta_{ij}. \tag{32}$$

Therefore, the propagator is readily obtained as follows

$$G_k = (\mathcal{P}_k)^{-1} = \begin{pmatrix} G_k^K & G_k^R \\ G_k^A & 0 \end{pmatrix}, \tag{33}$$

where the retarded and advanced components read

$$G_k^R = (\mathcal{P}_k^R)^{-1}, \qquad G_k^A = (\mathcal{P}_k^A)^{-1}, \tag{34}$$

and the Keldysh propagator or the correlation function is given by

$$G_k^K = -(\mathcal{P}_k^R)^{-1}\mathcal{P}_k^K(\mathcal{P}_k^A)^{-1} = -G_k^R \mathcal{P}_k^K G_k^A. \tag{35}$$

It is easy to verify a relation among the different components of the propagator as follows

$$G_k^K = \left(G_k^R - G_k^A\right)\coth\left(\frac{q_0}{2T}\right), \tag{36}$$

which is the fluctuation-dissipation relation in thermal equilibrium.

To summarize, the retarded, advanced, correlation (Keldysh) two-point connected Green's functions or propagators are given by

$$iG_{\sigma,k}^R = \langle T_p \sigma_c(x)\sigma_q(y)\rangle, \quad iG_{\sigma,k}^A = \langle T_p \sigma_q(x)\sigma_c(y)\rangle, \tag{37}$$

$$iG_{\sigma,k}^K = \langle T_p \sigma_c(x)\sigma_c(y)\rangle = \left(iG_k^R\right)\left(i\mathcal{P}_k^K\right)\left(iG_k^A\right), \tag{38}$$

for the $\sigma$ meson, and

$$i\left(G_{\pi,k}^R\right)_{ij} = \langle T_p \pi_{i,c}(x)\pi_{j,q}(y)\rangle, \tag{39}$$

$$i\left(G_{\pi,k}^A\right)_{ij} = \langle T_p \pi_{i,q}(x)\pi_{j,c}(y)\rangle, \tag{40}$$

$$i\left(G_{\pi,k}^K\right)_{ij} = \langle T_p \pi_{i,c}(x)\pi_{j,c}(y)\rangle, \tag{41}$$

for the $\pi$ meson. Here, $T_p$ is the time ordering operator in the closed time path from the positive branch to the negative one, and $\langle\cdots\rangle$ denotes ensemble average. Note that the last equality in Equation (38) results from Equation (35). In Figure 1 we show the diagrammatic representation for the retarded, advanced, and Keldysh propagators in this work. The retarded propagator is denoted by a dashed line with two points labelled with "$c,q$", and the advanced propagator with "$q,c$". Motivated by Equation (38), we use a line with an empty circle in its middle to represent the Keldysh propagator. Hence, the Keldysh propagator is essentially composed of the retarded and advanced propagators jointed with an empty circle, which corresponds to $i\mathcal{P}_k^K$ in Equation (38).

## 3 Flow of the effective potential

Inserting Equation (24) into Equation (22) and expanding the r.h.s. of Equation (22) in powers of $\mathcal{F}_k$, one is led to

$$\partial_\tau \Gamma_k[\Phi] = \frac{i}{2}\mathrm{STr}\left[\tilde{\partial}_\tau \ln G_k^{-1}\right] + \frac{i}{2}\mathrm{STr}\left[\tilde{\partial}_\tau\left(G_k\mathcal{F}_k\right)\right] - \frac{i}{4}\mathrm{STr}\left[\tilde{\partial}_\tau\left(G_k\mathcal{F}_k\right)^2\right]$$

$$+ \frac{i}{6}\mathrm{STr}\left[\tilde{\partial}_\tau\left(G_k\mathcal{F}_k\right)^3\right] - \frac{i}{8}\mathrm{STr}\left[\tilde{\partial}_\tau\left(G_k\mathcal{F}_k\right)^4\right] + \cdots, \tag{42}$$

which allows us to obtain the flow equations for various $n$-point Green's functions, e.g., the masses, wave function renormalization, couplings, etc., as shown in Equation (7).

The first term on the r.h.s. of Equation (42) is vanishing, which is straightforwardly verified by plugging in Equation (33) and Equation (20), i.e.,

$$\mathrm{STr}\left[\tilde{\partial}_\tau \ln G_k^{-1}\right] = \mathrm{STr}\left[\left(\partial_\tau \hat{R}_{\phi,k}\right)G_k^R\right] + \mathrm{STr}\left[\left(\partial_\tau \hat{R}_{\phi,k}\right)G_k^A\right]. \tag{43}$$

Figure 2: Diagrammatic representation of the flow equation for the effective potential. The external leg with a label "$q$" stands for the field $\sigma_q$, and the internal lines are the Keldysh propagators for the $\sigma$ and $\pi$. The gray blobs denote full vertices, and the crossed circles indicate the regulator insertion as shown in Equation (113).

And one has

$$\mathrm{STr}\!\left[\left(\partial_\tau \hat{R}_{\phi,k}\right)G_k^R\right] = \mathrm{STr}\!\left[\left(\partial_\tau \hat{R}_{\phi,k}\right)G_k^A\right] = 0\,, \tag{44}$$

which results from the fact that the retarded and advanced propagators are analytic in the upper or lower half of the complex plane in $q_0$, respectively, as shown in Equation (27) and Equation (28).

Let us proceed to the second term on the r.h.s. of Equation (42). After a simple calculation, one arrives at

$$\mathrm{STr}\!\left(G_k \mathcal{F}_k\right) = \int_x \left[-\lambda_{3\sigma,k} G_{\sigma,k}^K(x,x)\sigma_q(x) - \lambda_{1\sigma2\pi,k}\left(G_{\pi,k}^K\right)_{ii}(x,x)\sigma_q(x) + \cdots\right]. \tag{45}$$

Note that only terms relevant in the following are shown explicitly in Equation (45). Performing the projection as follows,

$$\partial_\tau \left(\frac{i\delta\Gamma_k[\Phi]}{\delta\sigma_q}\right)\Bigg|_{\Phi=0} = \frac{i}{2}\,\tilde{\partial}_\tau \left(\frac{i\delta\,\mathrm{STr}\!\left(G_k\mathcal{F}_k\right)}{\delta\sigma_q}\right)\Bigg|_{\Phi=0} + \cdots\,, \tag{46}$$

one is led to

$$\partial_\tau\!\left(-i\lambda_{1\sigma,k}\right) = \frac{1}{2}\tilde{\partial}_\tau \left[\left(-i\lambda_{3\sigma,k}\right)\!\left(iG_{\sigma,k}^K\right) + \left(-i\lambda_{1\sigma2\pi,k}\right)\!\left(iG_{\pi,k}^K\right)_{ii}\right], \tag{47}$$

which is depicted in Figure 2. Here the partial operator $\tilde{\partial}_\tau$ only hits the RG scale dependence through the regulator in propagators, which leaves us with the regulator insertion as shown in the second line of Figure 2. Note that, the regulator insertion takes place on either side of the Keldysh propagator, separated by the open circle.

The blobs in Figure 2 stands for the one-particle-irreducible (1PI) vertices, which are defined as

$$i\Gamma_{k,\Phi_{a_1}\cdots\Phi_{a_n}}^{(n)} = \left(\frac{i\delta^n\Gamma_k[\Phi]}{\delta\Phi_{a_1}\cdots\delta\Phi_{a_n}}\right)\Bigg|_{\Phi=0}\,, \tag{48}$$

for a general $n$-point function. Note that $\Phi_a$ in Equation (48) includes both the "classical" and "quantum" fields, which are distinguished in diagrams by a label "$c$" or "$q$" attached for

each external line of the vertices, as shown in Figure 2. Substituting Equation (15) into Equation (47), and employing the relations as follows

$$\lambda_{3\sigma,k} = \bar{\rho}_c^{\frac{1}{2}} \frac{\partial m_{\sigma,k}^2}{\partial \bar{\rho}_c}, \qquad \lambda_{1\sigma 2\pi,k} = \bar{\rho}_c^{\frac{1}{2}} \frac{\partial m_{\pi,k}^2}{\partial \bar{\rho}_c}, \tag{49}$$

one is led to

$$\partial_\tau V_k'(\bar{\rho}_c) = \frac{i}{4} \int \frac{d^4 q}{(2\pi)^4} \tilde{\partial}_\tau \Big[ \frac{\partial m_{\sigma,k}^2}{\partial \bar{\rho}_c} G_{\sigma,k}^K(q) + \frac{\partial m_{\pi,k}^2}{\partial \bar{\rho}_c} (G_{\pi,k}^K)_{ii}(q) \Big]$$

$$= \frac{\partial}{\partial \bar{\rho}_c} \Big\{ -\frac{i}{4} \int \frac{d^4 q}{(2\pi)^4} (\partial_\tau R_{\phi,k}(q)) \big[ G_{\sigma,k}^K(q) + (G_{\pi,k}^K)_{ii}(q) \big] \Big\}. \tag{50}$$

Thus, one is allowed to integrate both sides of equation above, which yields

$$\partial_\tau V_k(\bar{\rho}_c) = -\frac{i}{4} \int \frac{d^4 q}{(2\pi)^4} (\partial_\tau R_{\phi,k}(q)) \big[ G_{\sigma,k}^K(q) + (G_{\pi,k}^K)_{ii}(q) \big], \tag{51}$$

up to a term independent of $\bar{\rho}_c$. If Equation (27) and Equation (28) are used, one arrives at

$$\partial_\tau V_k(\bar{\rho}_c) = \frac{k^4}{4\pi^2} \Big[ l_0^{(B,4)}(\tilde{m}_{\sigma,k}^2, \eta_{\phi,k}; T) + (N-1) l_0^{(B,4)}(\tilde{m}_{\pi,k}^2, \eta_{\phi,k}; T) \Big], \tag{52}$$

with the RG invariant dimensionless meson masses $\tilde{m}_{\sigma,k}^2 = m_{\sigma,k}^2/(k^2 Z_{\phi,k})$, $\tilde{m}_{\pi,k}^2 = m_{\pi,k}^2/(k^2 Z_{\phi,k})$, and the anomalous dimension which is defined as follows

$$\eta_{\phi,k} \equiv -\frac{\partial_\tau Z_{\phi,k}}{Z_{\phi,k}}. \tag{53}$$

The threshold function in Equation (52) reads

$$l_0^{(B,4)}(\tilde{m}_{\phi,k}^2, \eta_{\phi,k}; T) = \frac{2}{3} \Big( 1 - \frac{\eta_{\phi,k}}{5} \Big) \frac{1}{\sqrt{1 + \tilde{m}_{\phi,k}^2}} \Big( \frac{1}{2} + n_B(\tilde{m}_{\phi,k}^2; T) \Big), \tag{54}$$

with the bosonic distribution function

$$n_B(\tilde{m}_{\phi,k}^2; T) = \frac{1}{\exp\Big\{ \frac{k}{T} \sqrt{1 + \tilde{m}_{\phi,k}^2} \Big\} - 1}. \tag{55}$$

Note that Equation (52) is nothing but the flow equation for the effective potential in the local potential approximation with an additional wave function renormalization, cf., e.g., [36, 46].

## 4 Flows of propagators and vertices

To proceed, we make projections for both sides of Equation (42), onto the inverse retarded propagators for the $\sigma$- and $\pi$-fields, respectively, i.e., $i\Gamma_{\sigma_q \sigma_c}^{(2)}$ and $i\Gamma_{\pi_{i,q} \pi_{j,c}}^{(2)}$. Then one is left with the flow equations for the inverse retarded $\sigma$- and $\pi$-propagators, as shown in Figure 3. Note that the flow equations in Figure 3 are the general ones, which are independent of truncations used, such as the LPA with a wave function renormalization in Equation (7); for example, the full vertices denoted by gray blobs could be momentum-dependent, whereas they are not in LPA.

$$\partial_\tau \left( \overset{q}{-\!-\!-\!\bullet\!-\!-\!} \overset{c}{-\!-\!} \right) = \tilde\partial_\tau \left( \frac{1}{2} \underset{q\ \ c}{\overset{q\ \ q}{\bigcirc}} + \frac{1}{2} \underset{q\ \ c}{\overset{q\ q}{\bigcirc}} \right.$$

$$\left. + \overset{q}{-\!-\!-\!\bullet} \overset{c\ \ c}{\underset{c}{\bigcirc}} \overset{c}{\bullet\!-\!-\!} + \overset{q}{-\!-\!-\!\bullet} \overset{c\ \ c}{\underset{c}{\bigcirc}} \overset{c}{\bullet\!-\!-\!} \right)$$

$$\partial_\tau \left( \overset{q}{-\!-\!-\!\bullet\!-\!-\!} \overset{c}{-\!-\!} \right) = \tilde\partial_\tau \left( \frac{1}{2} \underset{q\ \ c}{\overset{q\ \ q}{\bigcirc}} + \frac{1}{2} \underset{q\ \ c}{\overset{q\ q}{\bigcirc}} \right.$$

$$\left. + \overset{q}{-\!-\!-\!\bullet} \overset{c\ \ c}{\underset{c}{\bigcirc}} \overset{c}{\bullet\!-\!-\!} + \overset{q}{-\!-\!-\!\bullet} \overset{c\ \ c}{\underset{c}{\bigcirc}} \overset{c}{\bullet\!-\!-\!} \right)$$

Figure 3: Diagrammatic representation of the flow equations for the inverse retarded propagators, i.e., $i\Gamma^{(2)}_{\sigma_q \sigma_c}$ and $i\Gamma^{(2)}_{\pi_{i,q} \pi_{j,c}}$, and see Equation (48) for the definition.

$$\partial_\tau \left( \underset{q\ \ c}{\overset{c\ \ c}{\times\!\!\!\!\bigcirc\!\!\!\!\times}} \right) = \tilde\partial_\tau \left( \underset{q\ \ c}{\overset{c\ c}{\times\!\!\!\bigcirc\!\!\!\times}} + \underset{c\ \ c}{\overset{c}{\times\!\!\!\bigcirc\!\!\!\times}} + \underset{q\ c}{\overset{c\ c}{\bigcirc}} \right)$$

Figure 4: Diagrammatic representation of the flow equation for the four-point vertex $i\Gamma^{(4)}_{\phi_q \phi_c \phi_c \phi_c}$ in the symmetric phase.

In the following we focus on the case of the symmetric phase, i.e., the expected value of $\bar\phi_{0,c}$ in Equation (4) is vanishing, and thus the sigma and pion fields are degenerate, which will be denoted collectively with $\phi_i$ ($i = 0, 1, ...N-1$). In such case, the flow equation of the four-point vertex $i\Gamma^{(4)}_{\phi_q \phi_c \phi_c \phi_c}$ is given in Figure 4, where contributions from three-point vertices, e.g., those in Figure 3, are absent.

Due to the interchange symmetry for the external legs of the four-point vertex in the l.h.s. of flow equation in Figure 4, i.e.,

$$i\Gamma^{(4)}_{k,\phi_{i,q} \phi_{j,c} \phi_{k,c} \phi_{l,c}}(p_i, p_j, p_k, p_l) \equiv \quad \tag{56}$$

the four-point vertex could be parametrized generically as follows

$$i\Gamma^{(4)}_{k,\phi_{i,q} \phi_{j,c} \phi_{k,c} \phi_{l,c}}(p_i, p_j, p_k, p_l) = -\frac{i}{3} \Big[ \lambda^{\text{eff}}_{4\pi,k}(p_i, p_j, p_k, p_l)\delta_{il}\delta_{jk} + \lambda^{\text{eff}}_{4\pi,k}(p_i, p_k, p_l, p_j)\delta_{ij}\delta_{kl}$$

$$+ \lambda^{\text{eff}}_{4\pi,k}(p_i, p_l, p_j, p_k)\delta_{ik}\delta_{jl} \Big], \tag{57}$$

where we have introduced an effective four-point coupling $\lambda_{4\pi,k}^{\text{eff}}$, that is dependent on external momenta.

Apparently, the flow equation in Figure 4 is a self-consistent functional equation for the vertex, as same as the propagators in Figure 3. This functional differential equation, however, can be simplified significantly, once the requirement of the self-consistency is loosened a bit. For example, one could insert the vertices and propagators in LPA as in Equation (7) into the r.h.s. of the flow equation in Figure 4, and consequently, the one-loop vertex in, e.g., the $t$ channel reads

$$-iV_{ijkl}^{\text{loop}}(p) \equiv \qquad\qquad\qquad\qquad\qquad\qquad \tag{58}$$

with

$$-iV_{ijkl}^{\text{loop}}(p) = \int \frac{d^4q}{(2\pi)^4} \frac{-i\lambda_{4\pi,k}}{3}(\delta_{ii'}\delta_{ll'} + \delta_{il}\delta_{i'l'} + \delta_{il'}\delta_{li'})$$

$$\times i\left(G_{\pi,k}^K\right)_{l'k'}(q)\, i\left(G_{\pi,k}^A\right)_{j'i'}(q-p) \times \frac{-i\lambda_{4\pi,k}}{3}(\delta_{jj'}\delta_{kk'} + \delta_{jk'}\delta_{kj'} + \delta_{jk}\delta_{j'k'})$$

$$= (-i)\frac{\lambda_{4\pi,k}^2}{9}\Big[2(\delta_{ij}\delta_{kl} + \delta_{ik}\delta_{jl} + \delta_{il}\delta_{jk}) + \delta_{il}\delta_{jk}(N+2)\Big]I_k(p), \tag{59}$$

where we have defined a function as follows

$$I_k(p) \equiv i\int \frac{d^4q}{(2\pi)^4} G_{\pi,k}^K(q) G_{\pi,k}^A(q-p), \tag{60}$$

which receives contributions from both the real and imaginary parts, i.e.,

$$I_k(p) = \Re I_k(p) + i\Im I_k(p), \tag{61}$$

whose properties have been discussed in detail in Appendix B, and one can also find the explicit expressions therein.

Substituting Equation (57) and Equation (59) into the l.h.s. and r.h.s. of the flow equation in Figure 4, respectively, one is led to the flow equation for the effective four-point coupling, i.e.,

$$\partial_\tau \lambda_{4\pi,k}^{\text{eff}}(p_i, p_j, p_k, p_l) = \frac{\lambda_{4\pi,k}^2}{3}\Big[(N+4)\tilde\partial_\tau I_k(-p_i-p_l) + 2\,\tilde\partial_\tau I_k(-p_i-p_k) + 2\,\tilde\partial_\tau I_k(-p_i-p_j)\Big]. \tag{62}$$

Note that the computation of $\tilde\partial_\tau I_k(p)$ can be simplified, if the wave function renormalization $Z_{\phi,k}$, as shown in Equation (17), is assumed to be $Z_{\phi,k} = 1$, which is adopted in our numerical calculations for the r.h.s. of flow equations. Then one has

$$\tilde\partial_\tau I_k(p) = \partial_\tau\Big|_{\bar m_{\pi,k}^2} I_k(p), \tag{63}$$

with the $\bar m_{\pi,k}^2$ fixed. The r.h.s. of equation above can be calculated directly by resorting to the explicit expression of $I_k(p)$ in Appendix B.

With the momentum-dependent four-point vertex in Equation (57), one is allowed to construct the self-energy in the symmetric phase, which reads

$$-i\Sigma_{k,ij}(p) \equiv \frac{1}{2}$$

(64)

with

$$-i\Sigma_{k,ij}(p) = \delta_{ij}\left(-\frac{i}{6}\right)(N+2)\int \frac{d^4q}{(2\pi)^4} iG^K_{\pi,k}(q)\bar{\lambda}^{\text{eff}}_{4\pi,k}(p_0,|\boldsymbol{p}|,q_0,|\boldsymbol{q}|,\cos\theta),$$

(65)

where we have defined a function $\bar{\lambda}^{\text{eff}}_{4\pi,k}$, which reads

$$\bar{\lambda}^{\text{eff}}_{4\pi,k}(p_0,|\boldsymbol{p}|,q_0,|\boldsymbol{q}|,\cos\theta) = \frac{1}{N+2}\Big[N\lambda^{\text{eff}}_{4\pi,k}(-p,-q,q,p) + \lambda^{\text{eff}}_{4\pi,k}(-p,p,-q,q)$$

$$+ \lambda^{\text{eff}}_{4\pi,k}(-p,q,p,-q)\Big],$$

(66)

with the angle $\theta$ between the vectors $\boldsymbol{p}$ and $\boldsymbol{q}$. Then the flow of the inverse retarded propagator, i.e.,

$$i\Gamma^{(2)}_{k,\phi_{i,q}\phi_{j,c}}(p) \equiv i\frac{\delta^2\Gamma_k[\phi]}{\delta\phi_{i,q}(p)\delta\phi_{j,c}(-p)} = i\delta_{ij}\big(Z_{\phi,k}(p^2)p^2 - m^2_{\pi,k}\big),$$

(67)

is given by

$$\partial_\tau\Gamma^{(2)}_{k,\phi_q\phi_c}(p) = -\tilde{\partial}_\tau\Sigma_k(p) = \left(-\frac{i}{6}\right)(N+2)\int \frac{d^4q}{(2\pi)^4}\tilde{\partial}_\tau\big(G^K_{\pi,k}(q)\big)\bar{\lambda}^{\text{eff}}_{4\pi,k}(p_0,|\boldsymbol{p}|,q_0,|\boldsymbol{q}|,\cos\theta).$$

(68)

Inserting the Keldysh propagator in Equation (35) with Equation (27) and Equation (28) into the equation above, one is led to

$$\partial_\tau\Gamma^{(2)}_{k,\phi_q\phi_c}(p_0,|\boldsymbol{p}|) = \partial_\tau\Gamma^{(2)\text{I}}_{k,\phi_q\phi_c}(p_0,|\boldsymbol{p}|) + \partial_\tau\Gamma^{(2)\text{II}}_{k,\phi_q\phi_c}(p_0,|\boldsymbol{p}|),$$

(69)

with

$$\partial_\tau\Gamma^{(2)\text{I}}_{k,\phi_q\phi_c}(p_0,|\boldsymbol{p}|)) = -\frac{1}{24}\frac{(N+2)}{(2\pi)^2}\left[-\frac{\coth\left(\frac{E_{\pi,k}(k)}{2T}\right)}{\big(E_{\pi,k}(k)\big)^3} - \frac{\text{csch}^2\left(\frac{E_{\pi,k}(k)}{2T}\right)}{2T\big(E_{\pi,k}(k)\big)^2}\right]$$

$$\times (2k^2)\int_0^k d|\boldsymbol{q}||\boldsymbol{q}|^2\int_{-1}^1 d\cos\theta\Big[\bar{\lambda}^{\text{eff}}_{4\pi,k}\big|_{q_0=E_{\pi,k}(k)}$$

$$+ \bar{\lambda}^{\text{eff}}_{4\pi,k}\big|_{q_0=-E_{\pi,k}(k)}\Big],$$

(70)

and

$$\partial_\tau\Gamma^{(2)\text{II}}_{k,\phi_q\phi_c}(p_0,|\boldsymbol{p}|) = -\frac{1}{24}\frac{(N+2)}{(2\pi)^2}\frac{\coth\left(\frac{E_{\pi,k}(k)}{2T}\right)}{\big(E_{\pi,k}(k)\big)^2}(2k^2)\int_0^k d|\boldsymbol{q}||\boldsymbol{q}|^2$$

$$\times \int_{-1}^1 d\cos\theta\Big[\frac{\partial}{\partial q_0}\bar{\lambda}^{\text{eff}}_{4\pi,k}\big|_{q_0=E_{\pi,k}(k)} - \frac{\partial}{\partial q_0}\bar{\lambda}^{\text{eff}}_{4\pi,k}\big|_{q_0=-E_{\pi,k}(k)}\Big],$$

(71)

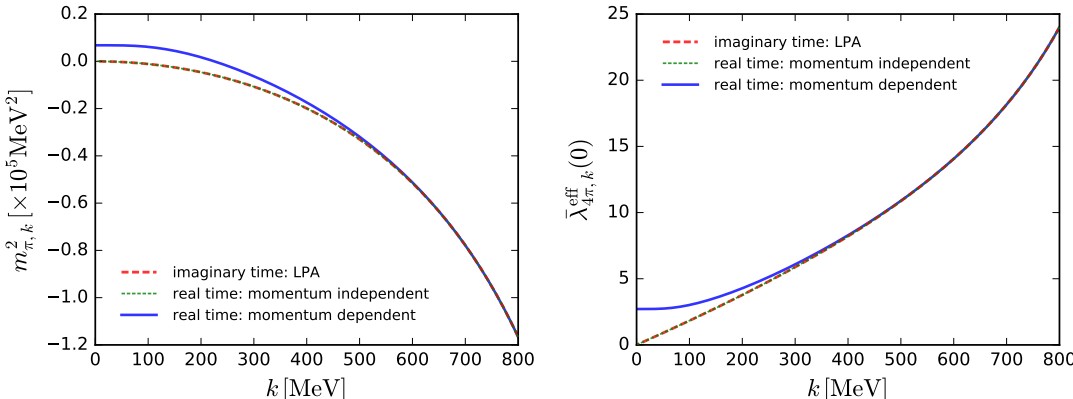

Figure 5: Meson mass square of the effective potential $m^2_{\pi,k}$ (left panel) in Equation (9) as well as Equation (67) and the effective four-meson coupling $\bar{\lambda}^{\text{eff}}_{4\pi,k}(0)$ with vanishing momenta (right panel) in Equation (66) as functions of the RG scale $k$ in the $O(4)$ scalar theory with temperature $T = 145\,\text{MeV}$ and initial values of the flow equations at UV cutoff $\Lambda = 800\,\text{MeV}$ as follows: $m^2_{\pi,k=\Lambda} = -(341.45\,\text{MeV})^2$ and $\lambda_{4\pi,k=\Lambda} = 24$, where the expectation value of the field $\bar{\phi}_c$ in Equation (4) is chosen to be vanishing. The red dashed lines denote the results obtained in LPA calculation in the imaginary-time formalism, while the blue solid and green dashed lines stand for those obtained in the real-time formalism, with or without the momentum dependence of the effective four-meson coupling $\bar{\lambda}^{\text{eff}}_{4\pi,k}$ in the self-energy in Equation (64) included, respectively. Note that in order to facilitate the comparison, the contribution from six-point correlations to the flow of four-point vertex in the imaginary-time formalism is neglected.

where we have divided the flow of two-point correlation function into two parts, denoted by superscripts I and II, respectively. One can see that the second part in Equation (71) arises from the derivative of vertex w.r.t. $q_0$, which can be neglected if the momentum dependence of the vertex is mild. In Equation (70) and Equation (71) one has

$$E_{\pi,k}(k) = \left(k^2 + m^2_{\pi,k}\right)^{1/2}. \tag{72}$$

## 5 Numerical results

We have set up the flow equations for the two- and four-point correlation functions in the section above, cf. Equation (68) and Equation (62), respectively. It is desirable to compare calculated results in this formalism to those in conventional Euclidean formalism in some limiting cases, which allows us to verify the correctness of formalism of fRG within the Keldysh field theory. For instance, in Equation (62) if the external momentum dependence of the effective four-point coupling is ignored, to be identified with $\lambda_{4\pi,k}$ on the r.h.s., to wit,

$$\lambda^{\text{eff}}_{4\pi,k}(0) = \lambda_{4\pi,k}, \tag{73}$$

then the flow equations of $\lambda_{4\pi,k}$ in Equation (62) and $m^2_{\pi,k}$ extracted in Equation (68) with vanishing momenta, i.e.,

$$\Gamma^{(2)}_{k,\phi_q\phi_c}(0) = -m^2_{\pi,k}, \tag{74}$$

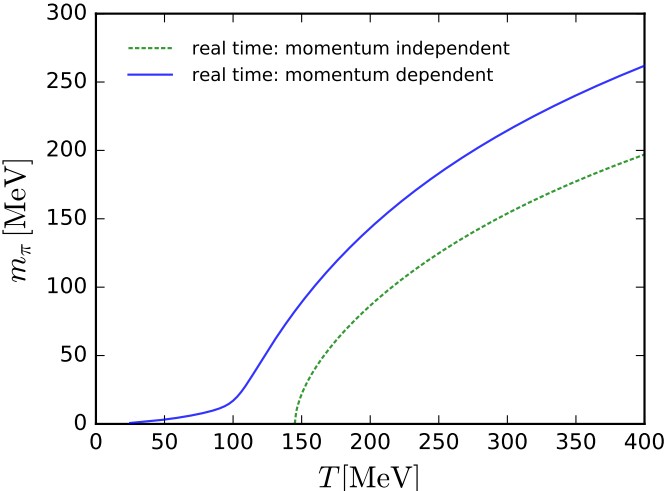

Figure 6: Mass in the infrared limit $m_{\pi,k=0}$ as a function of temperature. The initial conditions for the flows are given by the UV cutoff $\Lambda = 800\,\text{MeV}$, $m_{\pi,k=\Lambda}^2 = -(341.45\,\text{MeV})^2$, and $\lambda_{4\pi,k=\Lambda} = 24$. The two different lines denote those obtained in the real-time formalism, with or without the momentum dependence of the effective four-meson coupling $\bar{\lambda}_{4\pi,k}^{\text{eff}}$ in the self-energy in Equation (64) included, respectively.

constitute a close set of equations, which can be solved self-consistently. This truncation described above is essentially the local potential approximation, and the calculated results should be identical to the relevant results in Equation (52), where the flow equation of the effective potential can be solved in the Euclidean spacetime, and the mass and coupling are obtained as derivatives of the effective potential w.r.t. the field, as shown in Equation (9) and Equation (13).

In Figure 5 we show the running of the meson mass square of the effective potential and the four-meson coupling with the RG scale. Note that in this work we focus on the temperature regime of $T \gtrsim T_c$, where $T_c$ is the critical temperature for the phase transition and the $O(4)$ symmetry is restored above $T_c$. Hence, the expectation value of the classical field $\bar{\phi}_c$ in Equation (4) is chosen to be vanishing here as well as in what follows. In Figure 5 we compare the calculations both from the real-time and imaginary-time formalisms, where the momentum dependence of the four-point vertex and the wave function renormalization for the propagator are not taken into account for the red and green dashed lines. Obviously, one observes that these two lines agree with each other exactly both for the mass and coupling, which indicates that the real-time fRG flows in this work are correct. Furthermore, we also perform the calculation with the momentum dependence of the effective four-meson coupling $\bar{\lambda}_{4\pi,k}^{\text{eff}}$ included in the self-energy in Equation (64), and the relevant results are shown in Figure 5 in blue solid lines. One can see that the effect of momentum dependence of the vertex plays an increasing role with the decrease of the RG scale. Note that the effective potential is broken in the ultraviolet, and the curvature of potential at $\phi_c = 0$, i.e., the squared meson mass, cf. Equation (9), is negative when the temperature is below $T_c$. When the temperature is increased above $T_c$, the squared meson mass evolves from the negative to a positive value with the decrease of the RG scale $k$. Therefore, the critical temperature just corresponds to the case that the meson mass square is vanishing at $k \to 0$.

For the two different real-time truncations, we also show their respective scalar mass in the IR limit $k \to 0$ as a function of the temperature in Figure 6. One can see that with the

same initial conditions and temperature, the mass is larger for the momentum dependent calculation. When the temperature is decreased down to $T = 145$ MeV, the mass is vanishing for the momentum independent calculation, and thus the critical temperature in this case is $T_c = 145$ MeV. However, we find a kink in the line of mass obtained in the momentum dependent truncation at about $T = 100$ MeV, as shown in Figure 6, and the relevant mass approaches zero when the temperature is at $T_c = 20.4$ MeV. In the following we will employ the momentum dependent truncation with the same initial conditions as used in Figure 6, otherwise stated explicitly.

It is interesting to explore underlying reasons accounting for the difference between the momentum independent and dependent results. When the momentum dependence is included, the flow of the effective four-point coupling in Equation (62) is suppressed at finite external momenta. Consequently, the coupling in the flow of the inverse retarded propagator in Equation (70), which contributes mostly around $|\boldsymbol{q}| \sim k$ due to the 3-$d$ momentum integral, is relatively larger than that for the case without momentum dependence. The larger coupling leads to an increased flow of the two-point function as well as a larger meson mass square $m_{\pi,k}^2$ in the infrared, as shown by the blue solid line in the left panel of Figure 5. Hence, lower temperature is required to decrease $m_{\pi,k}^2$ at $k \to 0$ in order to realize the phase transition. Furthermore, it is found that the kink-like structure of the blue line around about $T = 100$ MeV in Figure 6 arises from the fact that when the temperature is below $\sim 100$ MeV, the meson mass square in the region of low $k$ behaves as $m_{\pi,k}^2 \sim -k^2$, which results in a small energy factor $E_{\pi,k}(k)$ in Equation (70), cf. also Equation (72), and eventually increases the flow of the inverse retarded propagator even further. That is the reason why the critical temperature in the case with momentum dependence is significantly lower than that without momentum dependence.

## 5.1 Imaginary parts of the vertex and inverse retarded propagator

We proceed with discussing the imaginary part of the inverse retarded propagator in Equation (68). As shown in Equation (70) and Equation (71), the imaginary part of $\Gamma_k^{(2)}$ arises from the imaginary part of the effective four-point vertex $\bar{\lambda}_{4\pi,k}^{\text{eff}}$ in Equation (66). As we have discussed above, when the momentum dependence of the vertex is mild, the contribution in Equation (71) can be neglected. In this work in order to simplify numerical calculations, we refrain from taking Equation (71) into account, and hope to report its contribution in the near future.

Inspired by the equation in Equation (70), one defines the internal momentum $\boldsymbol{q}$ averaged effective vertex, as follows

$$\tilde{\lambda}_{4\pi,k}^{\text{eff}}(p_0, |\boldsymbol{p}|) = \frac{3}{4k^3} \int_0^k d|\boldsymbol{q}||\boldsymbol{q}|^2 \int_{-1}^1 d\cos\theta \left[ \bar{\lambda}_{4\pi,k}^{\text{eff}}\big|_{q_0 = E_{\pi,k}(k)} + \bar{\lambda}_{4\pi,k}^{\text{eff}}\big|_{q_0 = -E_{\pi,k}(k)} \right]. \tag{75}$$

In Figure 7 we show the dependence of the imaginary part of $-\tilde{\lambda}_{4\pi,k}^{\text{eff}}$ on $p_0$ with $|\boldsymbol{p}| = 0$ at several different values of RG scale $k$. Note that the imaginary part is vanishing at the UV cutoff $k = \Lambda$. One can see that the imaginary part of the averaged effective vertex is vanishing as $p_0 \to 0$ as it should, since it is an odd function with $p_0 \to -p_0$. An interesting result is that, the imaginary part of $-\tilde{\lambda}_{4\pi,k}^{\text{eff}}$ is negative in the regime of small $p_0$, which is more obvious in the inlay. Moreover, when $p_0$ is increased up to $E_{\pi,k}$, denoted by the positions of dots in Figure 7, it jumps to a positive value. This behavior is due to the function $I_{1,k}(p)$ as shown in Equation (117), which is responsible for creation and annihilation of particles, and the kinematic window is open when $p_0$ is larger than $E_{\pi,k}$.

In order to explore the underlying reason for the negative value of the imaginary part of $-\tilde{\lambda}_{4\pi,k}^{\text{eff}}$ in the regime of small $p_0$ as shown in Figure 7, we plug Equation (66) into Equa-

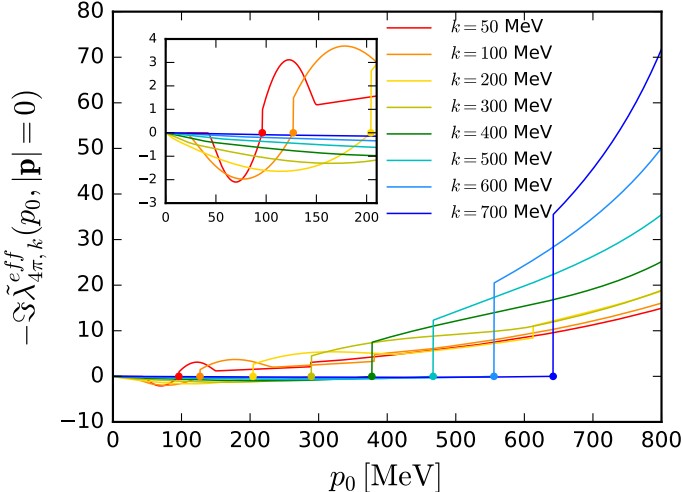

Figure 7: Imaginary part of $-\tilde{\lambda}_{4\pi,k}^{\text{eff}}(p_0,|\boldsymbol{p}|)$ defined in Equation (75) as a function of $p_0$ with temperature $T = 145$ MeV, where the spacial momentum is chosen to be vanishing. The mass square $m_{\pi,k}^2$ in Equation (28) and $\lambda_{4\pi,k}$ in Equation (62) are input from the results of self-consistent computation including the momentum dependence as shown by the solid blue lines in Figure 5. Different lines correspond to difference values of the RG scale $k$. The inlay shows the zoomed-in view in the region of small $p_0$.

tion (62), and arrive at

$$\partial_\tau \bar{\lambda}_{4\pi,k}^{\text{eff}}(p_0,|\boldsymbol{p}|,q_0,|\boldsymbol{q}|,\cos\theta) = \frac{\lambda_{4\pi,k}^2}{3}\Big[(N+2)\tilde{\partial}_\tau I_k(0) + 3\tilde{\partial}_\tau I_k(p-q) + 3\tilde{\partial}_\tau I_k(p+q)\Big]. \quad (76)$$

Apparently, the first term in the square bracket on the r.h.s. does not contribute to the imaginary part. Thus, we only need to focus on the other two terms. Moreover, we divide $I_k = I_{1,k} + I_{2,k}$, see Equation (114). As shown in Equation (117) and Equation (118) in Appendix B, the function $I_{1,k}$ corresponds to the on-shell creation and annihilation of two particles, and $I_{2,k}$ describes the process of particles scattering in the heat bath, i.e., Landau damping. Note that only $I_{1,k}$ receives a contribution in vacuum, and $I_{2,k}$ is vanishing at $T = 0$. We show different contributions arising from different parts to the imaginary part of $-\tilde{\lambda}_{4\pi,k}^{\text{eff}}$ in Figure 8. The left and right panels correspond to two values of the RG scale, $k = 100, 600$ MeV, respectively. One observes that when $k$ is large, the result is dominated by $I_{1,k}$, since the thermal effect is negligible at large $k$. With the decrease of RG scale, contributions from both $I_{1,k}$ and $I_{2,k}$ are comparable to each other, as shown in the left panel of Figure 8. Note that results from $I_{1,k}$ are always nonnegative and their values are positive once $p_0$ is above some threshold values. On the contrary, result of $I_{2,k}$ is negative at small $p_0$, and it crosses zero and changes sign with increasing $p_0$. Finally, it vanishes at large $p_0$. To summarize, it is found that the negative value of the imaginary part of $-\tilde{\lambda}_{4\pi,k}^{\text{eff}}$ at small $p_0$ is due to the Landau damping.

In Figure 9 we show the dependence of the imaginary part of two-point correlation function $\Gamma_{\phi_q \phi_c}^{(2)}(p_0,|\boldsymbol{p}|=0)$ on the temporal momentum $p_0$ with $T = 145$ MeV. Different contributions from $I_{1,k}$ and $I_{2,k}$ are also depicted. As shown in the solid blue line in Figure 6, the meson mass in the infrared is finite at $T = 145$ MeV, about 100 MeV, so $\Im\,\Gamma_{\phi_q\phi_c}^{(2)}$ from $I_{1,k}$ is significantly suppressed in the region of small $p_0$, while it is dominated by Landau damping, i.e., $I_{2,k}$. As a consequence, the total imaginary part of the two-point correlation function shown in Figure 9

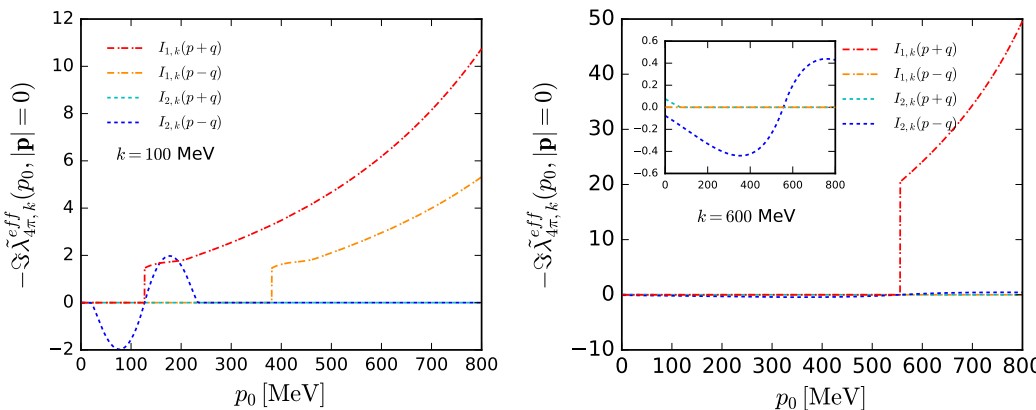

Figure 8: Different parts of contribution to the imaginary part of $-\tilde{\lambda}^{\text{eff}}_{4\pi,k}(p_0,|\boldsymbol{p}|=0)$ as a function of $p_0$ with temperature $T = 145$ MeV at $k = 100$ MeV (left panel) and $k = 600$ MeV (right panel). We also show the zoomed-in view in the inlay in the right panel.

is negative when $p_0$ is small, and increases and becomes positive when $p_0$ is large, where the particle creation and annihilation take over the relevant dynamics.

## 5.2 Spectral functions

In the Källén-Lehmann spectral representation, the retarded propagator in Equation (34) reads

$$G_R(p_0,|\boldsymbol{p}|) = -\int_{-\infty}^{\infty} \frac{dp_0'}{2\pi} \frac{\rho(p_0',|\boldsymbol{p}|)}{p_0'-(p_0+i\epsilon)}\,, \tag{77}$$

where the RG scale is assumed to be in the IR limit $k \to 0$, and $\rho$ on the r.h.s. is the spectral function. It is straightforwardly to obtain the relation between the spectral function and the imaginary part of retarded propagator, to wit,

$$\rho(p_0,|\boldsymbol{p}|) = -2\Im\, G_R(p_0,|\boldsymbol{p}|)\,. \tag{78}$$

Moreover, one also has

$$G_R(p_0,|\boldsymbol{p}|) = \left[\Gamma^{(2)}_{\phi_q\phi_c}(p_0,|\boldsymbol{p}|)\right]^{-1}\,, \tag{79}$$

which led us to the expression for the spectral function, as follows

$$\rho(p_0,|\boldsymbol{p}|) = \frac{2\Im\,\Gamma^{(2)}_{\phi_q\phi_c}(p_0,|\boldsymbol{p}|)}{\left[\Re\,\Gamma^{(2)}_{\phi_q\phi_c}(p_0,|\boldsymbol{p}|)\right]^2 + \left[\Im\,\Gamma^{(2)}_{\phi_q\phi_c}(p_0,|\boldsymbol{p}|)\right]^2}\,. \tag{80}$$

Obviously, the spectral function is an odd function of $p_0$, viz.

$$\rho(-p_0,|\boldsymbol{p}|) = -\rho(p_0,|\boldsymbol{p}|)\,. \tag{81}$$

In Figure 10 we show the 3D plots of the imaginary and real parts of the two-point correlation function $\Gamma^{(2)}_{\phi_q\phi_c}(p_0,|\boldsymbol{p}|)$ as functions of $p_0$ and $|\boldsymbol{p}|$ with $T = 145$ MeV. The gray planes are the zero planes with $z = 0$, which intersect with the surface of the two-point correlation function at a dashed curve colored in red. One can observe obviously that the imaginary part

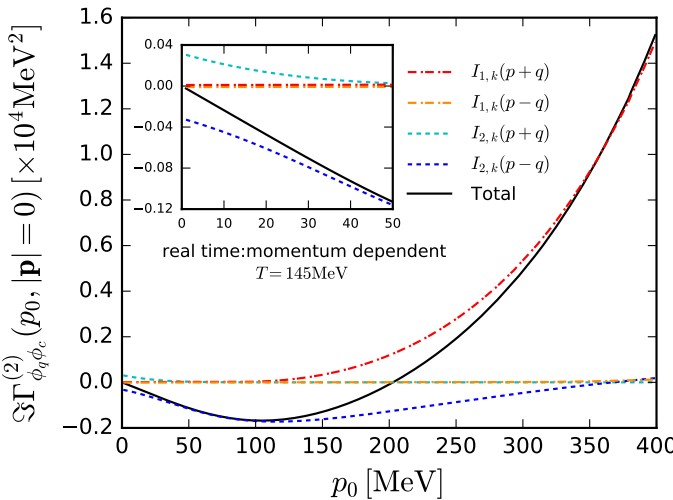

Figure 9: Imaginary part of the two-point correlation function $\Gamma^{(2)}_{\phi_q\phi_c}(p_0, |\boldsymbol{p}| = 0)$ in the IR limit $k \to 0$ as a function of $p_0$ at temperature $T = 145$ MeV. Different contributions and the total one are shown, respectively. The inlay shows the zoomed-in view in the region of small $p_0$.

of the two-point correlation function, i.e., the inverse retarded propagator, is below the zero plane in a regime of small $p_0$, and this regime grows a bit with the increasing magnitude of spacial momentum. As we have discussed in Section 5.1 in detail, the negative imaginary part is due to the Landau damping. Moreover, one can find the real part of the inverse retarded propagator on the right panel of Figure 10 is also below the zero plane. This is because the temperature here is $T = 145$ MeV, which is above the relevant critical value $T_c = 20.4$ MeV, as shown by the blue curve in Figure 6, and thus mass square is positive, cf. Equation (67).

In Figure 11 we show the imaginary and real parts of the inverse retarded propagator $\Gamma^{(2)}_{\phi_q\phi_c}(p_0, |\boldsymbol{p}| = 0)$ as a function of $p_0$ with several values of temperature. An interesting finding is that with the decrease of temperature, when the temperature is below about 60 MeV, the mass is very small as shown by the blue solid line in Figure 6, the process of creation and annihilation of particles governed by $I_{1,k}$ in Equation (114) dominates over the Landau damping by $I_{2,k}$. As a consequence, the negative imaginary part in the regime of small $p_0$ disappears and becomes positive, as shown in the inlay of the left panel of Figure 11. One can see this more clearly in Figure 12, where the spectral function $\rho(p_0, |\boldsymbol{p}| = 0)$ is depicted as a function of $p_0$ with different values of temperature. One observes that when the temperature is large, as shown in the right panel of Figure 12, the spectral function is negative in the region of small $p_0$ and a minus peak structure develops around a pole mass. However, when the temperature is below about 60 MeV as shown in the left panel of Figure 12, the spectral function is positive in the whole region of positive $p_0$, and the peak becomes more and more wider and finally disappears as the temperature is approaching the critical value. Furthermore, we have inspected the process $\phi \to 3\phi$ in the spectral function in Figure 12. This process can be traced back to the imaginary part of the internal momentum averaged effective vertex in Figure 8, where in the left panel one can see that the sudden rise of the threshold function $I_{1,k}(p-q)$ just corresponds to $p_0 = 3E_{\pi,k}$. However, its contribution to the spectral function is almost hidden by processes related to, e.g., $I_{1,k}(p+q)$ and $I_{2,k}(p-q)$ as shown in Figure 9, and hence the process $\phi \to 3\phi$ is hard to be observed from the spectral function in Figure 12.

As we have demonstrated above, when the temperature is above and not far away from

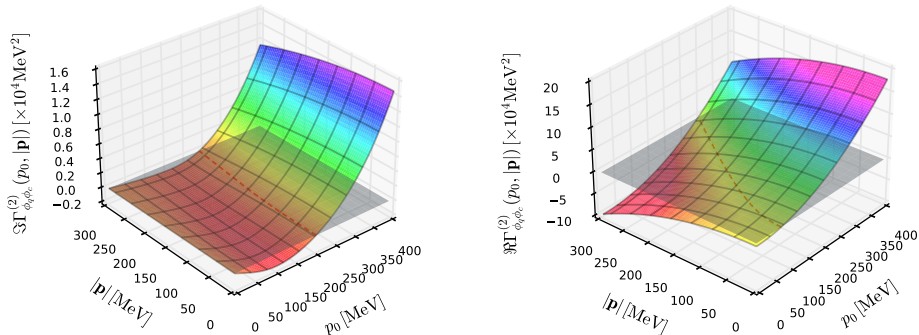

Figure 10: 3D plots of the imaginary (left panel) and real (right panel) parts of the two-point correlation function $\Gamma^{(2)}_{\phi_q \phi_c}(p_0, |\boldsymbol{p}|)$ as functions of $p_0$ and $|\boldsymbol{p}|$ with temperature $T = 145$ MeV. Here the zero planes, i.e., those with the $z$-axis $z = 0$, are also shown to guide the eyes. The two surfaces intersect with each other at the red dashed line.

the critical temperature, the spectral function is positive. However, it is found that when the temperature is quite larger than the critical one, Landau damping contributes a negative value to the spectral function in the regime of small $p_0$. Whether it is an artifact in our computation certainly needs more sophisticated investigations. For instance, the truncation for the flow equation of the four-point vertex as shown in Figure 4 might have to be improved in the region of high temperature, where the momentum dependence should also be encoded for the four-point vertices on the r.h.s. of the flow equation in Figure 4. We hope to report the relevant studies in future work.

In both Figure 11 and Figure 12, we have used the solid lines to denote the results of low temperature and the dashed lines those of high temperature. We close this subsection with Figure 13, in which the 3D plots of the spectral function as a function of $p_0$ and $|\boldsymbol{p}|$ with a low temperature $T = 54$ MeV and a high one $T = 145$ MeV are presented. The ridge structure related to the peak in Figure 12 is obvious in both 3D plots, and for the high temperature, it even crosses from the negative region to the positive one.

## 5.3 Dynamical critical exponent

The kinetic coefficient $\Gamma(|\boldsymbol{p}|)$ can be defined as

$$\frac{1}{\Gamma(|\boldsymbol{p}|)} = -i \left. \frac{\partial \Gamma^{(2)}_{\phi_q \phi_c}(p_0, |\boldsymbol{p}|)}{\partial p_0} \right|_{p_0=0} = \left. \frac{\partial \Im \Gamma^{(2)}_{\phi_q \phi_c}(p_0, |\boldsymbol{p}|)}{\partial p_0} \right|_{p_0=0}, \tag{82}$$

where we have used the fact as follows

$$\Re \Gamma^{(2)}_{\phi_q \phi_c}(-p_0, |\boldsymbol{p}|) = \Re \Gamma^{(2)}_{\phi_q \phi_c}(p_0, |\boldsymbol{p}|), \tag{83}$$

$$\Im \Gamma^{(2)}_{\phi_q \phi_c}(-p_0, |\boldsymbol{p}|) = -\Im \Gamma^{(2)}_{\phi_q \phi_c}(p_0, |\boldsymbol{p}|). \tag{84}$$

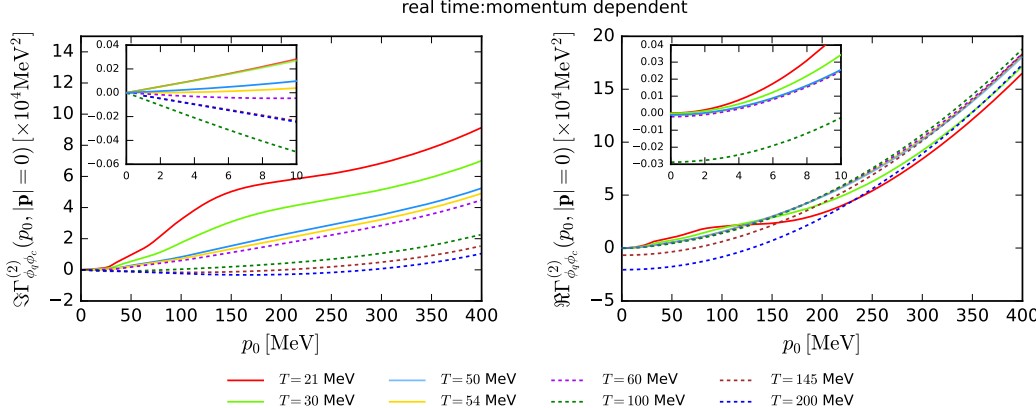

Figure 11: Imaginary (left panel) and real (right panel) parts of the two-point correlation function $\Gamma^{(2)}_{\phi_q\phi_c}(p_0, |\boldsymbol{p}| = 0)$ as functions of $p_0$ with several values of temperature. The inlays show the zoomed-in view in the region of small $p_0$.

Consequently, the real part does not contribute to Equation (82). The dissipative characteristic frequency, or relaxation rate, reads

$$\omega(|\boldsymbol{p}|) = \Gamma(|\boldsymbol{p}|)\left(-\Gamma^{(2)}_{\phi_q\phi_c}(p_0 = 0, |\boldsymbol{p}|)\right)$$

$$= -\Gamma(|\boldsymbol{p}|)\Re\,\Gamma^{(2)}_{\phi_q\phi_c}(p_0 = 0, |\boldsymbol{p}|). \tag{85}$$

The relaxation rate varies as

$$\omega(|\boldsymbol{p}|) \propto |\boldsymbol{p}|^z, \tag{86}$$

when $|\boldsymbol{p}| > \xi^{-1}$, with the correlation length $\xi \sim m_\phi^{-1}$ [14]. The power $z$ in Equation (86) is the dynamical critical exponent. In Figure 14 we depict the double logarithm plot of the relaxation rate $\omega(|\boldsymbol{p}|)$ in Equation (86) as a function of the spacial momentum with temperature $T = T_c = 20.4$ MeV. From this plot one can extract the value of the dynamical critical exponent, and we obtain $z = 2.02284(6)$, where only the numerical or statistical error, rather than the systematic one, is included.

Interestingly, this value of the dynamical critical exponent obtained in this work is compatible with a very recent result for Model A in three spatial dimensions, $z = 1.92(11)$, obtained from real-time classical-statistical lattice simulations [76]. Here we have used the standard classification for the universality of critical dynamics [14]. However, the critical dynamics of the relativistic $O(4)$ scalar theory should be more closely related to Model G, based on the analysis by Rajagopal and Wilczek [77], see also [78]. The dynamical critical exponent of Model G is $z = 3/2$ in three dimensions. Though direct calculation of the dynamical critical exponent for the $O(4)$ model from classical-statistical lattice simulations has not yet arrived at a conclusive result because of errors, it indicates that $z$ is in favor of 2 [78]. Furthermore, it is also found that the dynamical critical exponent in a relativistic $O(N)$ vector model is close to 2 [63]. Similar result is also found for the $O(3)$ model in [75]. In summary, whether the critical dynamics of the relativistic $O(4)$ scalar theory falls into Model A or Model G is still an open question, and more insightful studies are required.

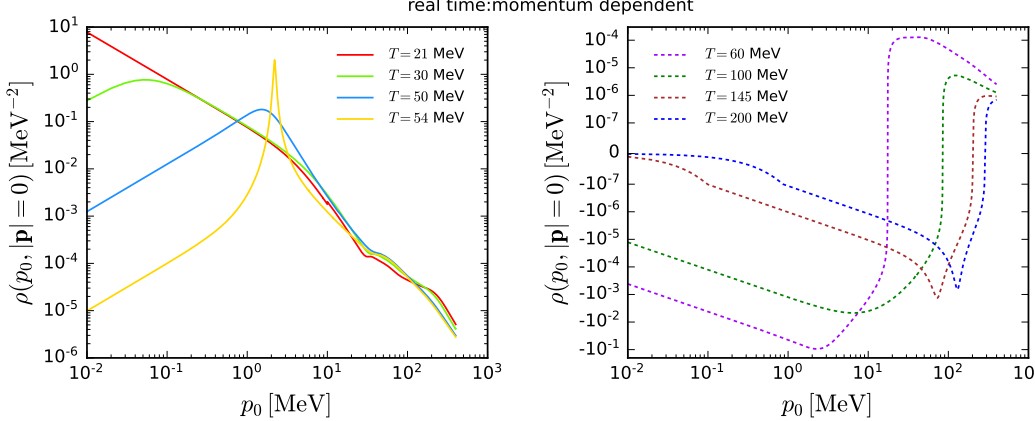

Figure 12: Spectral function $\rho(p_0, |\boldsymbol{p}| = 0)$ as a function of $p_0$ with several small (left panel) and large (right panel) values of temperature. In the right panel, a symmetric log scale is applied for the $y$-axis in order to take into account both positive and negative values of the spectral function, where the log scale is implicitly translated into a linear one upon crossing the zero point.

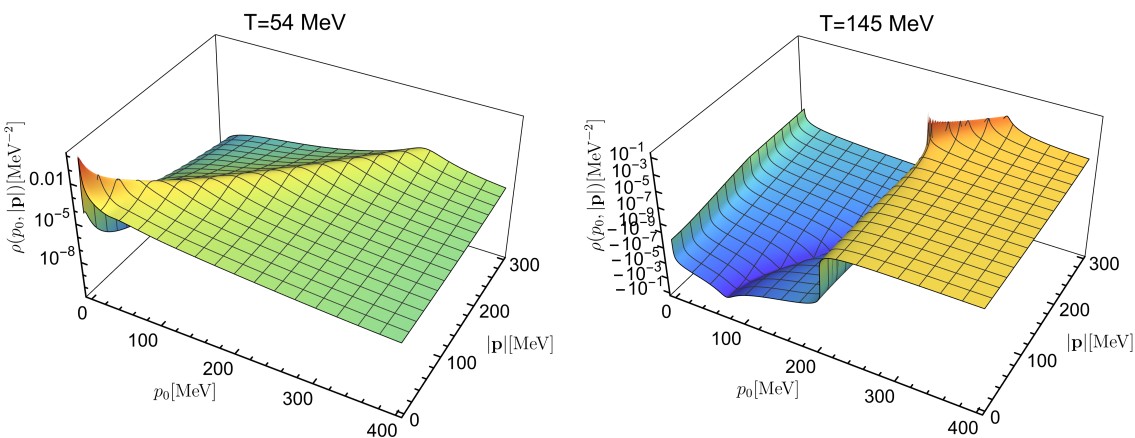

Figure 13: 3D plots of the spectral function as a function of $p_0$ and $|\boldsymbol{p}|$ with temperature $T = 54$ MeV (left panel) and 145 MeV (right panel).

# 6 Conclusions

In this work we have studied the real-time dynamics of the $O(4)$ scalar theory within the functional renormalization group formulated on the Schwinger-Keldysh closed time path. The effective action and flow equations are organized in terms of two classes of fields, i.e., the "classical" and "quantum" fields, which in this work are denoted by subscripts $c$ and $q$, respectively. A concise diagrammatic representation for the propagators, including the retarded, advanced and Keldysh propagators, and various vertices are introduced and used in the derivation of flow equations. We have demonstrated in detail that this formalism in the real-time fRG produces identical results for the effective potential, meson mass, and four-point vertex, in comparison to the relevant results obtained in the imaginary-time fRG, when the momentum dependence of vertices is suppressed.

We have solved the flow equations for the momentum-dependent two- and four-point correlation functions in the symmetric phase at finite temperature. A simplified self-consistent truncation has been used, which allows us to obtain analytic expressions for the flows of the

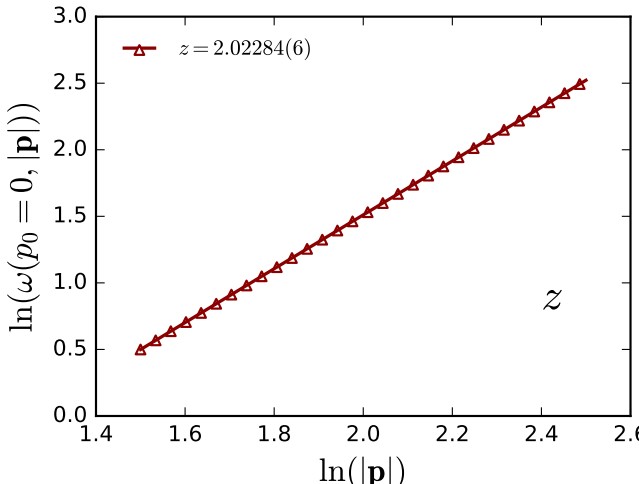

Figure 14: Double logarithm plot of the dissipative characteristic frequency $\omega(|\boldsymbol{p}|)$ in Equation (86) as a function of the spacial momentum with temperature $T = T_c = 20.4$ MeV.

propagator and vertex. We have investigated in detail roles of two different processes, i.e., the on-shell creation and annihilation of particles and Landau damping, in the imaginary parts of the two- and four-point correlation functions, as well as in the spectral function. We find that Landau damping probably leads to negative spectral functions and imaginary part of correlation functions at high temperature, which is certainly required to be confirmed in future studies with more improved truncation. Spectral functions with different values of temperature and spacial momentum are obtained. Moreover, we have calculated the dynamical critical exponent for the phase transition near the critical temperature in the $O(4)$ scalar theory in 3+1 dimensions, and found $z \simeq 2.023$.

We have to point out that the computation in this work is not fully self-consistent, and Equation (71) is also neglected in the calculation of the inverse retarded propagator, which certainly should be improved in future work. Even for that, we have shown in this work that the combination between the fRG and the Keldysh functional integral is promising, and it is able to provide us with a wealth of insights on the real-time dynamics of nonperturbative field theories. Therefore, it is very interesting to apply this formalism in more realistic theory, e.g., the Yang-Mills theory, which we hope to report in near future.

# Acknowledgements

We thank Jens Braun, Jan Horak, Jia-sen Jin, Jan M. Pawlowski, Fabian Rennecke, Nicolas Wink and Yue-Liang Wu for illuminating discussions. This work is supported by the National Natural Science Foundation of China under Grant No. 11775041 and by the Fundamental Research Funds for the Central Universities under Grant No. DUT20GJ212. W.F. also would like to acknowledge the support from the Peng Huanwu Visiting Professor Program For Young Scientists during his visiting at Institute of Theoretical Physics, Chinese Academy of Sciences.

# A  Formalism of the fRG in the Keldysh field theory

Given a collective notation for all the fields concerned, $\varphi = \{\varphi_i\}$, where the subscript $i$ not only distinguishes different species of fields, but also denotes the space-time coordinates and other internal degrees of freedom, the classical Keldysh action for a closed system reads

$$S[\varphi] = \int_x \Big( \mathcal{L}[\varphi_+] - \mathcal{L}[\varphi_-] \Big), \tag{87}$$

with the shorthand notation $\int_x \equiv \int_{-\infty}^{\infty} dt \int d^3x$, where $\mathcal{L}$ is a generic Lagrangian density, and $\varphi_\pm$ stand for the fields on the forward and backward branches, respectively. This also applies for variables with indices $\pm$ in what follows. The Keldysh generating functional is given by

$$Z[J_+, J_-] = \int \big( \mathcal{D}\varphi_+ \mathcal{D}\varphi_- \big) \exp\Big\{ i\Big( S[\varphi] + \big( J_+^i \varphi_{i,+} - J_-^i \varphi_{i,-} \big) \Big) \Big\}, \tag{88}$$

where $J_\pm^i$ are the external sources conjugate to $\varphi_{i,\pm}$, respectively. Note that summations and/or integrals are assumed for repeated indices. By the use of the Keldysh rotation as follows

$$\begin{cases} \varphi_{i,+} = \frac{1}{\sqrt{2}}(\varphi_{i,c} + \varphi_{i,q}), \\[2mm] \varphi_{i,-} = \frac{1}{\sqrt{2}}(\varphi_{i,c} - \varphi_{i,q}), \end{cases} \tag{89}$$

and

$$\begin{cases} J_+^i = \frac{1}{\sqrt{2}}(J_c^i + J_q^i), \\[2mm] J_-^i = \frac{1}{\sqrt{2}}(J_c^i - J_q^i), \end{cases} \tag{90}$$

where quantities with the subscripts $c$ and $q$ stand for physical "classical" and "quantum" variables, respectively, one is able to reformulate Equation (88) in terms of physical fields and external sources, to wit,

$$Z[J_c, J_q] = \int \big( \mathcal{D}\varphi_c \mathcal{D}\varphi_q \big) \exp\Big\{ i\Big( S[\varphi] + \big( J_q^i \varphi_{i,c} + J_c^i \varphi_{i,q} \big) \Big) \Big\}. \tag{91}$$

Then, we employ the two-point regulator term as follows

$$\Delta S_k[\varphi] = \frac{1}{2}(\varphi_{i,c}, \varphi_{i,q}) \begin{pmatrix} 0 & R_k^{ij} \\ (R_k^{ij})^* & 0 \end{pmatrix} \begin{pmatrix} \varphi_{j,c} \\ \varphi_{j,q} \end{pmatrix} = \frac{1}{2}\Big( \varphi_{i,c} R_k^{ij} \varphi_{j,q} + \varphi_{i,q} (R_k^{ij})^* \varphi_{j,c} \Big), \tag{92}$$

where the IR regulator, viz.,

$$R_k^{ij} = \begin{cases} R_k^{ji} & \text{bosonic} \\[3mm] -R_k^{ji} & \text{fermionic} \end{cases}, \tag{93}$$

is used to suppress quantum fluctuations of momenta less than a RG scale $k$, i.e., $q \lesssim k$. Inserting Equation (92) into Equation (91), one is left with the scale-dependent generating functional, which reads

$$Z_k[J_c, J_q] = \int \big( \mathcal{D}\varphi_c \mathcal{D}\varphi_q \big) \exp\Big\{ i\Big( S[\varphi] + \Delta S_k[\varphi] + \big( J_q^i \varphi_{i,c} + J_c^i \varphi_{i,q} \big) \Big) \Big\}. \tag{94}$$

It follows that the generating functional for the connected Green's functions, i.e., Schwinger functional, is also $k$-dependent, i.e.,

$$W_k[J_c, J_q] = -i \ln Z_k[J_c, J_q]. \tag{95}$$

In the following , we would like to absorb the suffixes $c$ and $q$ into the index $i$, and denote them collectively with a new label $a$, i.e.,

$$\{\varphi_a\} = \left\{ \{\varphi_{i,c}\}, \{\varphi_{i,q}\} \right\}, \tag{96}$$

$$\{J^a\} = \left\{ \{J_q^i\}, \{J_c^i\} \right\}. \tag{97}$$

Hence, the expectation value of field $\varphi_a$ is given by

$$\Phi_a \equiv \langle \varphi_a \rangle = \frac{\delta W_k[J]}{\delta J^a}, \tag{98}$$

and the two-point connected Green's function, i.e., the propagator, reads

$$G_{k,ab} \equiv -i \langle \varphi_a \varphi_b \rangle_c = -i \left[ \langle \varphi_a \varphi_b \rangle - \langle \varphi_a \rangle \langle \varphi_b \rangle \right] = -\frac{\delta^2 W_k[J]}{\delta J^a J^b}. \tag{99}$$

In the same way, other notations can be simplified by the new label, and for instance, the regulator term in Equation (92) then reads

$$\Delta S_k[\varphi] = \frac{1}{2} \varphi_a R_k^{ab} \varphi_b, \tag{100}$$

with

$$R_k^{ab} \equiv \begin{pmatrix} 0 & R_k^{ij} \\ (R_k^{ij})^* & 0 \end{pmatrix}. \tag{101}$$

The Legendre transformation of Schwinger functional $W_k[J]$ allows us to obtain the one-particle-irreducible (1PI) effective action, given by

$$\Gamma_k[\Phi] = W_k[J] - \Delta S_k[\Phi] - J^a \Phi_a. \tag{102}$$

Introducing a symbol as follows

$$\gamma^a{}_b = \begin{cases} -\delta^a{}_b & \text{$a$ and $b$ are fermionic} \\ \delta^a{}_b & \text{others} \end{cases}, \tag{103}$$

one thus has

$$J^a \Phi_a = \gamma^a{}_b \Phi_a J^b. \tag{104}$$

Differentiating both sides of Equation (102) with respect to $\Phi_a$ and employing Equation (104), one is led to

$$\frac{\delta(\Gamma_k[\Phi] + \Delta S_k[\Phi])}{\delta \Phi_a} = -\gamma^a{}_b J^b, \tag{105}$$

i.e.,

$$\frac{\delta^2(\Gamma_k[\Phi] + \Delta S_k[\Phi])}{\delta J^b \delta \Phi_a} = -\gamma^a{}_b. \tag{106}$$

The l.h.s. of Equation (106) can be further reformulated as

$$\frac{\delta^2(\Gamma_k[\Phi] + \Delta S_k[\Phi])}{\delta J^b \delta \Phi_a} = \frac{\delta \Phi_c}{\delta J^b} \frac{\delta^2(\Gamma_k[\Phi] + \Delta S_k[\Phi])}{\delta \Phi_c \delta \Phi_a} = \frac{\delta^2 W_k[J]}{\delta J^b \delta J^c} \frac{\delta^2(\Gamma_k[\Phi] + \Delta S_k[\Phi])}{\delta \Phi_c \delta \Phi_a}$$

$$= - G_{k,bc} \frac{\delta^2(\Gamma_k[\Phi] + \Delta S_k[\Phi])}{\delta \Phi_c \delta \Phi_a}, \tag{107}$$

where we have used Equation (99). Thus, one arrives at

$$G_{k,bc} \frac{\delta^2(\Gamma_k[\Phi] + \Delta S_k[\Phi])}{\delta \Phi_c \delta \Phi_a} = \gamma^a{}_b. \tag{108}$$

Employing the notation as follows

$$\left(\Gamma_k^{(2)}[\Phi] + \Delta S_k^{(2)}[\Phi]\right)^{ab} \equiv \frac{\delta^2(\Gamma_k[\Phi] + \Delta S_k[\Phi])}{\delta \Phi_a \delta \Phi_b}, \tag{109}$$

one can obtain the propagator from Equation (108), which reads

$$G_{k,ab} = \gamma^c{}_a \left[\left(\Gamma_k^{(2)}[\Phi] + \Delta S_k^{(2)}[\Phi]\right)^{-1}\right]_{cb}. \tag{110}$$

Let us proceed to considering the flow equation for Schwinger functional $W_k[J]$, which is readily obtained by differentiating both sides of Equation (95) with respect to the RG time $\tau \equiv \ln(k/\Lambda)$, with an initial evolution scale $\Lambda$, i.e., the UV cutoff. One arrives at

$$\partial_\tau W_k[J] = \frac{1}{Z_k[J]} \int \mathcal{D}\varphi \left(\partial_\tau \Delta S_k[\varphi]\right) \exp\left\{i\left(S[\varphi] + \Delta S_k[\varphi] + J^a \varphi_a\right)\right\}$$

$$= \frac{1}{2} \frac{1}{Z_k[J]} \int \mathcal{D}\varphi \left(\varphi_a \partial_\tau R_k^{ab} \varphi_b\right) \exp\left\{i\left(S[\varphi] + \Delta S_k[\varphi] + J^a \varphi_a\right)\right\}$$

$$= \frac{1}{2} \langle \varphi_a \varphi_b \rangle \partial_\tau R_k^{ab}$$

$$= \frac{1}{2}(i G_{k,ab} + \Phi_a \Phi_b) \partial_\tau R_k^{ab}, \tag{111}$$

where we have used Equation (99) for the last equality. Given the property of the interchange of indices for the regulator in Equation (93) as well as the matrix form in Equation (101), the flow equation of Schwinger functional above can be reformulated slightly such that

$$\partial_\tau W_k[J] = \frac{i}{2} \text{STr}\left[\left(\partial_\tau R_k^*\right) G_k\right] + \frac{1}{2} \Phi_a \left(\partial_\tau R_k^{ab}\right) \Phi_b, \tag{112}$$

where the super trace, denoted by STr, provides an additional minus sign for the fermionic degrees of freedom. Finally, by the use of Equation (102), one is led to the flow equation for the effective action as follows

$$\partial_\tau \Gamma_k[\Phi] = \partial_\tau W_k[J] - \partial_\tau \Delta S_k[\Phi] = \frac{i}{2} \text{STr}\left[\left(\partial_\tau R_k^*\right) G_k\right]. \tag{113}$$

## B   Function $I_k(p)$ in Equation (60)

Plugging the propagators in Equation (34) and Equation (35) into Equation (60), and dividing it into two parts, one is led to

$$I_k(p) = I_{1,k}(p) + I_{2,k}(p), \tag{114}$$

with

$$I_{1,k}(p) = \Re I_{1,k}(p) + i\Im I_{1,k}(p), \tag{115}$$

$$I_{2,k}(p) = \Re I_{2,k}(p) + i\Im I_{2,k}(p), \tag{116}$$

where the imaginary parts reads

$$\Im I_{1,k}(p) = \frac{1}{Z_{\phi,k}^2}\left(-\frac{\pi}{4}\right)\int \frac{d^3q}{(2\pi)^3}\frac{1}{E_{\pi,k}(q)E_{\pi,k}(q-p)}$$

$$\times \coth\left(\frac{E_{\pi,k}(q)}{2T}\right)\left[\delta\left(E_{\pi,k}(q) + E_{\pi,k}(q-p) - p^0\right)\right.$$

$$\left. - \delta\left(-E_{\pi,k}(q) - E_{\pi,k}(q-p) - p^0\right)\right], \tag{117}$$

and

$$\Im I_{2,k}(p) = \frac{1}{Z_{\phi,k}^2}\left(-\frac{\pi}{4}\right)\int \frac{d^3q}{(2\pi)^3}\frac{1}{E_{\pi,k}(q)E_{\pi,k}(q-p)}$$

$$\times \coth\left(\frac{E_{\pi,k}(q)}{2T}\right)\left[\delta\left(-E_{\pi,k}(q) + E_{\pi,k}(q-p) - p^0\right)\right.$$

$$\left. - \delta\left(E_{\pi,k}(q) - E_{\pi,k}(q-p) - p^0\right)\right]. \tag{118}$$

Here one has

$$E_{\pi,k}(q) = \left[\boldsymbol{q}^2\left(1 + r_B\left(\frac{\boldsymbol{q}^2}{k^2}\right)\right) + \bar{m}_{\pi,k}^2\right]^{1/2}, \tag{119}$$

with $\bar{m}_{\pi,k}^2 = m_{\pi,k}^2/Z_{\phi,k}$. Note that $I_{1,k}(p)$ is related to creation and annihilation of particles, and $I_{2,k}(p)$ describes the Landau damping. From the expression of Equation (118), it is readily obtained that $\Im I_{2,k}$ is vanishing at $T = 0$.

   The real parts in Equation (115) and Equation (116) are related to the imaginary ones through principal value integrals as follows

$$\Re I_{1,k}(p) = \int_{-\infty}^{\infty} \frac{dp_0'}{\pi}\mathcal{P}\left(\frac{1}{p_0' - p_0}\right)\Im I_{1,k}(p_0', \boldsymbol{p}), \tag{120}$$

$$\Re I_{2,k}(p) = \int_{-\infty}^{\infty} \frac{dp_0'}{\pi}\mathcal{P}\left(\frac{1}{p_0' - p_0}\right)\Im I_{2,k}(p_0', \boldsymbol{p}), \tag{121}$$

where $\mathcal{P}$ denotes the principal values. From the expressions in Equation (117), Equation (118), Equation (120), and Equation (121), it is readily obtained that

$$\Im I_{1/2,k}(-p_0, \boldsymbol{p}) = -\Im I_{1/2,k}(p_0, \boldsymbol{p}), \tag{122}$$

$$\Re I_{1/2,k}(-p_0, \boldsymbol{p}) = \Re I_{1/2,k}(p_0, \boldsymbol{p}). \tag{123}$$

Although tedious, it is straightforward to perform the integrals in Equation (117) and Equation (118) for the explicit expressions of $\Im I_{1,k}(p)$ and $\Im I_{2,k}(p)$, and moreover, due to the property of odd function in Equation (122), it is only necessary to consider the case $p_0 \geq 0$. Prior to showing the results, it is more convenient to define several functions as follows

$$\mathcal{F}_1(q_+, q_-, p, \bar{m}_{\pi,k}^2) \equiv -\frac{1}{16\pi}\frac{1}{p}\left[\left(E(q_+, \bar{m}_{\pi,k}^2) - E(q_-, \bar{m}_{\pi,k}^2)\right) + 2T\ln\left(\frac{1 - e^{-E(q_+, \bar{m}_{\pi,k}^2)/T}}{1 - e^{-E(q_-, \bar{m}_{\pi,k}^2)/T}}\right)\right], \tag{124}$$

with

$$E(x, m^2) \equiv \sqrt{x^2 + m^2}. \tag{125}$$

The second function reads

$$\mathcal{F}_2(q_+, q_-, k, p, \bar{m}_{\pi,k}^2) \equiv -\frac{1}{16\pi}\frac{1}{p}\left[\frac{1}{2E(k, \bar{m}_{\pi,k}^2)}\left(k^2 - q_-^2\right)\coth\left(\frac{E(k, \bar{m}_{\pi,k}^2)}{2T}\right)\right.$$

$$\left. + \left(E(q_+, \bar{m}_{\pi,k}^2) - E(k, \bar{m}_{\pi,k}^2)\right) + 2T\ln\left(\frac{1 - e^{-E(q_+, \bar{m}_{\pi,k}^2)/T}}{1 - e^{-E(k, \bar{m}_{\pi,k}^2)/T}}\right)\right]. \tag{126}$$

And the third one is given by

$$\mathcal{F}_3(q_+, k, p, \bar{m}_{\pi,k}^2) \equiv -\frac{1}{16\pi}\frac{q_+}{E(k, \bar{m}_{\pi,k}^2)}\coth\left(\frac{E(q_+, \bar{m}_{\pi,k}^2)}{2T}\right)\left(1 - \frac{q_+^2 + p^2 - k^2}{2q_+ p}\right). \tag{127}$$

Moreover, we also need the counterparts of the three functions above with the vacuum contributions subtracted, which are given as follows

$$\mathcal{F}_1'(q_+, q_-, p, \bar{m}_{\pi,k}^2) \equiv -\frac{1}{16\pi}\frac{1}{p}\left[2T\ln\left(\frac{1 - e^{-E(q_+, \bar{m}_{\pi,k}^2)/T}}{1 - e^{-E(q_-, \bar{m}_{\pi,k}^2)/T}}\right)\right], \tag{128}$$

and

$$\mathcal{F}_2'(q_+, q_-, k, p, \bar{m}_{\pi,k}^2) \equiv -\frac{1}{16\pi}\frac{1}{p}\left[\frac{(k^2 - q_-^2)}{2E(k, \bar{m}_{\pi,k}^2)}\left(\coth\left(\frac{E(k, \bar{m}_{\pi,k}^2)}{2T}\right) - 1\right)\right.$$

$$\left. + 2T\ln\left(\frac{1 - e^{-E(q_+, \bar{m}_{\pi,k}^2)/T}}{1 - e^{-E(k, \bar{m}_{\pi,k}^2)/T}}\right)\right]. \tag{129}$$

The third one reads

$$\mathcal{F}_3'(q_+, k, p, \bar{m}_{\pi,k}^2) \equiv -\frac{1}{16\pi}\frac{q_+}{E(k, \bar{m}_{\pi,k}^2)}\left(\coth\left(\frac{E(q_+, \bar{m}_{\pi,k}^2)}{2T}\right) - 1\right)\left(1 - \frac{q_+^2 + p^2 - k^2}{2q_+ p}\right). \tag{130}$$

Moreover, we also need another function proportional to the delta function, which reads

$$\mathcal{F}_4(p_0, k, p, \bar{m}_{\pi,k}^2) \equiv -\frac{1}{16\pi} \frac{1}{\left(E(k, \bar{m}_{\pi,k}^2)\right)^2} \coth\left(\frac{E(k, \bar{m}_{\pi,k}^2)}{2T}\right)$$

$$\times \left(\frac{2}{3}k^3 - \frac{k^2 p}{2} + \frac{p^3}{24}\right) \delta\left(p_0 - 2E(k, \bar{m}_{\pi,k}^2)\right). \tag{131}$$

In the following, we show the explicit expressions of $\mathfrak{I}I_{1,k}(p)$ and $\mathfrak{I}I_{2,k}(p)$ in the formalism of piecewise functions, and the wave function renormalization is assumed to be $Z_{\phi,k} = 1$. Furthermore, two different cases of $\bar{m}_{\pi,k}^2 \geq 0$ and $\bar{m}_{\pi,k}^2 < 0$ are dealt with separately, which corresponds to the positive and negative curvatures of the potential in Equation (9), respectively. We begin with the case of $\bar{m}_{\pi,k}^2 \geq 0$.

I. If $k > |p|$, one has

(1). when $p^0 \geq (E(k + |p|, \bar{m}_{\pi,k}^2) + E(k, \bar{m}_{\pi,k}^2))$, then

$$\mathfrak{I}I_{1,k}(p) = \mathcal{F}_1(q_+, q_-, |p|, \bar{m}_{\pi,k}^2), \tag{132}$$

with

$$q_- = -\frac{|p|}{2} + \frac{\sqrt{p_0^2(p_0^2 - p^2)(p_0^2 - p^2 - 4\bar{m}_{\pi,k}^2)}}{2(p_0^2 - p^2)}, \tag{133}$$

$$q_+ = q_- + |p|; \tag{134}$$

(2). when $p^0 < (E(k + |p|, \bar{m}_{\pi,k}^2) + E(k, \bar{m}_{\pi,k}^2))$ and $p^0 > 2E(k, \bar{m}_{\pi,k}^2)$, then

$$\mathfrak{I}I_{1,k}(p) = \mathcal{F}_2(q_+, q_-, k, |p|, \bar{m}_{\pi,k}^2) + \mathcal{F}_3(q_+, k, |p|, \bar{m}_{\pi,k}^2), \tag{135}$$

with

$$q_- = \sqrt{(p^0 - E(k, \bar{m}_{\pi,k}^2))^2 - \bar{m}_{\pi,k}^2} - |p|, \tag{136}$$

$$q_+ = q_- + |p|; \tag{137}$$

(3). when $p^0 = 2E(k, \bar{m}_{\pi,k}^2)$, then

$$\mathfrak{I}I_{1,k}(p) = \mathcal{F}_4(p_0, k, |p|, \bar{m}_{\pi,k}^2); \tag{138}$$

(4). when $p^0 < 2E(k, \bar{m}_{\pi,k}^2)$, then

$$\mathfrak{I}I_{1,k}(p) = 0. \tag{139}$$

II. If $|p|/2 < k \leq |p|$, one has

(1). when $p^0 \geq (E(k + |p|, \bar{m}_{\pi,k}^2) + E(k, \bar{m}_{\pi,k}^2))$, then

$$\mathfrak{I}I_{1,k}(p) = \mathcal{F}_1(q_+, q_-, |p|, \bar{m}_{\pi,k}^2), \tag{140}$$

with

$$q_- = -\frac{|p|}{2} + \frac{\sqrt{p_0^2(p_0^2 - p^2)(p_0^2 - p^2 - 4\bar{m}_{\pi,k}^2)}}{2(p_0^2 - p^2)}, \tag{141}$$

$$q_+ = q_- + |p|; \tag{142}$$

(2). when $p^0 < \left(E(k+|\boldsymbol{p}|,\bar{m}^2_{\pi,k})+E(k,\bar{m}^2_{\pi,k})\right)$ and $p^0 \geq \left(E(|\boldsymbol{p}|,\bar{m}^2_{\pi,k})+E(k,\bar{m}^2_{\pi,k})\right)$, then

$$\Im I_{1,k}(p) = \mathcal{F}_2(q_+,q_-,k,|\boldsymbol{p}|,\bar{m}^2_{\pi,k})+\mathcal{F}_3(q_+,k,|\boldsymbol{p}|,\bar{m}^2_{\pi,k}), \tag{143}$$

with

$$q_- = \sqrt{(p^0-E(k,\bar{m}^2_{\pi,k}))^2-\bar{m}^2_{\pi,k}}-|\boldsymbol{p}|, \tag{144}$$

$$q_+ = q_- + |\boldsymbol{p}|; \tag{145}$$

(3). when $p^0 < \left(E(|\boldsymbol{p}|,\bar{m}^2_{\pi,k})+E(k,\bar{m}^2_{\pi,k})\right)$ and $p^0 > 2E(k,\bar{m}^2_{\pi,k})$, then

$$\Im I_{1,k}(p) = \mathcal{F}_2(q_+,q_-,k,|\boldsymbol{p}|,\bar{m}^2_{\pi,k})+\mathcal{F}_3(q_+,k,|\boldsymbol{p}|,\bar{m}^2_{\pi,k}), \tag{146}$$

with

$$q_- = |\boldsymbol{p}|-\sqrt{(p^0-E(k,\bar{m}^2_{\pi,k}))^2-\bar{m}^2_{\pi,k}}, \tag{147}$$

$$q_+ = \sqrt{(p^0-E(k,\bar{m}^2_{\pi,k}))^2-\bar{m}^2_{\pi,k}}; \tag{148}$$

(4). when $p^0 = 2E(k,\bar{m}^2_{\pi,k})$, then

$$\Im I_{1,k}(p) = \mathcal{F}_4(p_0,k,|\boldsymbol{p}|,\bar{m}^2_{\pi,k}); \tag{149}$$

(5). when $p^0 < 2E(k,\bar{m}^2_{\pi,k})$, then

$$\Im I_{1,k}(p) = 0. \tag{150}$$

III. If $k \leq |\boldsymbol{p}|/2$, one has

(1). when $p^0 \geq (E(k+|\boldsymbol{p}|,\bar{m}^2_{\pi,k})+E(k,\bar{m}^2_{\pi,k}))$, then

$$\Im I_{1,k}(p) = \mathcal{F}_1(q_+,q_-,|\boldsymbol{p}|,\bar{m}^2_{\pi,k}), \tag{151}$$

with

$$q_- = -\frac{|\boldsymbol{p}|}{2}+\frac{\sqrt{p_0^2(p_0^2-\boldsymbol{p}^2)(p_0^2-\boldsymbol{p}^2-4\bar{m}^2_{\pi,k})}}{2(p_0^2-\boldsymbol{p}^2)}, \tag{152}$$

$$q_+ = q_- + |\boldsymbol{p}|; \tag{153}$$

(2). when $p^0 < \left(E(k+|\boldsymbol{p}|,\bar{m}^2_{\pi,k})+E(k,\bar{m}^2_{\pi,k})\right)$ and $p^0 \geq \left(E(|\boldsymbol{p}|,\bar{m}^2_{\pi,k})+E(k,\bar{m}^2_{\pi,k})\right)$, then

$$\Im I_{1,k}(p) = \mathcal{F}_2(q_+,q_-,k,|\boldsymbol{p}|,\bar{m}^2_{\pi,k})+\mathcal{F}_3(q_+,k,|\boldsymbol{p}|,\bar{m}^2_{\pi,k}), \tag{154}$$

with

$$q_- = \sqrt{(p^0-E(k,\bar{m}^2_{\pi,k}))^2-\bar{m}^2_{\pi,k}}-|\boldsymbol{p}|, \tag{155}$$

$$q_+ = q_- + |\boldsymbol{p}|; \tag{156}$$

(3). when $p^0 < \left(E(|\boldsymbol{p}|, \bar{m}^2_{\pi,k}) + E(k, \bar{m}^2_{\pi,k})\right)$ and $p^0 \geq \left(E(k, \bar{m}^2_{\pi,k}) + E(|\boldsymbol{p}| - k, \bar{m}^2_{\pi,k})\right)$, then

$$\Im I_{1,k}(p) = \mathcal{F}_2(q_+, q_-, k, |\boldsymbol{p}|, \bar{m}^2_{\pi,k}) + \mathcal{F}_3(q_+, k, |\boldsymbol{p}|, \bar{m}^2_{\pi,k}),\tag{157}$$

with

$$q_- = |\boldsymbol{p}| - \sqrt{(p^0 - E(k, \bar{m}^2_{\pi,k}))^2 - \bar{m}^2_{\pi,k}},\tag{158}$$

$$q_+ = \sqrt{(p^0 - E(k, \bar{m}^2_{\pi,k}))^2 - \bar{m}^2_{\pi,k}};\tag{159}$$

(4). when $p^0 < \left(E(k, \bar{m}^2_{\pi,k}) + E(|\boldsymbol{p}| - k, \bar{m}^2_{\pi,k})\right)$ and $p^0 \geq 2E(|\boldsymbol{p}|/2, \bar{m}^2_{\pi,k})$, then

$$\Im I_{1,k}(p) = \mathcal{F}_1(q_+, q_-, |\boldsymbol{p}|, \bar{m}^2_{\pi,k}),\tag{160}$$

with

$$q_- = \frac{|\boldsymbol{p}|}{2} - \frac{\sqrt{p_0^2(p_0^2 - \boldsymbol{p}^2)(p_0^2 - \boldsymbol{p}^2 - 4\bar{m}^2_{\pi,k})}}{2(p_0^2 - \boldsymbol{p}^2)},\tag{161}$$

$$q_+ = \frac{|\boldsymbol{p}|}{2} + \frac{\sqrt{p_0^2(p_0^2 - \boldsymbol{p}^2)(p_0^2 - \boldsymbol{p}^2 - 4\bar{m}^2_{\pi,k})}}{2(p_0^2 - \boldsymbol{p}^2)};\tag{162}$$

(5). when $p^0 < 2E(|\boldsymbol{p}|/2, \bar{m}^2_{\pi,k})$, then

$$\Im I_{1,k}(p) = 0.\tag{163}$$

Next, we move on to the expression of $\Im I_{2,k}(p)$ with $\bar{m}^2_{\pi,k} \geq 0$, which reads as follows.

I. If $k > |\boldsymbol{p}|$, one has

(1). when $p^0 > |\boldsymbol{p}|$, then

$$\Im I_{2,k}(p) = 0,\tag{164}$$

(2). when $p^0 > \left(E(k + |\boldsymbol{p}|, \bar{m}^2_{\pi,k}) - E(k, \bar{m}^2_{\pi,k})\right)$ and $p^0 \leq |\boldsymbol{p}|$, then

$$\Im I_{2,k}(p) = \mathcal{F}'_1(q_+, q_-, |\boldsymbol{p}|, \bar{m}^2_{\pi,k}),\tag{165}$$

with

$$q_- = -\frac{|\boldsymbol{p}|}{2} - \frac{\sqrt{p_0^2(p_0^2 - \boldsymbol{p}^2)(p_0^2 - \boldsymbol{p}^2 - 4\bar{m}^2_{\pi,k})}}{2(p_0^2 - \boldsymbol{p}^2)},\tag{166}$$

$$q_+ = q_- + |\boldsymbol{p}|;\tag{167}$$

(3). when $p^0 \leq \left(E(k + |\boldsymbol{p}|, \bar{m}^2_{\pi,k}) - E(k, \bar{m}^2_{\pi,k})\right)$, then

$$\Im I_{2,k}(p) = \mathcal{F}'_2(q_+, q_-, k, |\boldsymbol{p}|, \bar{m}^2_{\pi,k}) - \mathcal{F}'_3(q_+, k, |\boldsymbol{p}|, \bar{m}^2_{\pi,k}),\tag{168}$$

with

$$q_- = \sqrt{(p^0 + E(k, \bar{m}^2_{\pi,k}))^2 - \bar{m}^2_{\pi,k}} - |\boldsymbol{p}|,\tag{169}$$

$$q_+ = q_- + |\boldsymbol{p}|.\tag{170}$$

II. If $|\boldsymbol{p}|/2 < k \le |\boldsymbol{p}|$, one has

    (1). when $p^0 > |\boldsymbol{p}|$, then

$$\Im I_{2,k}(p) = 0,\tag{171}$$

    (2). when $p^0 > \left(E(k+|\boldsymbol{p}|,\bar{m}^2_{\pi,k}) - E(k,\bar{m}^2_{\pi,k})\right)$ and $p^0 \le |\boldsymbol{p}|$, then

$$\Im I_{2,k}(p) = \mathcal{F}'_1(q_+,q_-,|\boldsymbol{p}|,\bar{m}^2_{\pi,k}),\tag{172}$$

    with

$$q_- = -\frac{|\boldsymbol{p}|}{2} - \frac{\sqrt{p_0^2(p_0^2-\boldsymbol{p}^2)(p_0^2-\boldsymbol{p}^2-4\bar{m}^2_{\pi,k})}}{2(p_0^2-\boldsymbol{p}^2)},\tag{173}$$

$$q_+ = q_- + |\boldsymbol{p}|;\tag{174}$$

    (3). when $p^0 > \left(E(|\boldsymbol{p}|,\bar{m}^2_{\pi,k}) - E(k,\bar{m}^2_{\pi,k})\right)$ and $p^0 \le \left(E(k+|\boldsymbol{p}|,\bar{m}^2_{\pi,k}) - E(k,\bar{m}^2_{\pi,k})\right)$, then

$$\Im I_{2,k}(p) = \mathcal{F}'_2(q_+,q_-,k,|\boldsymbol{p}|,\bar{m}^2_{\pi,k}) - \mathcal{F}'_3(q_+,k,|\boldsymbol{p}|,\bar{m}^2_{\pi,k}),$$

    with

$$q_- = \sqrt{(p^0 + E(k,\bar{m}^2_{\pi,k}))^2 - \bar{m}^2_{\pi,k}} - |\boldsymbol{p}|,\tag{175}$$

$$q_+ = q_- + |\boldsymbol{p}|;\tag{176}$$

    (4). when $p^0 \le \left(E(|\boldsymbol{p}|,\bar{m}^2_{\pi,k}) - E(k,\bar{m}^2_{\pi,k})\right)$, then

$$\Im I_{2,k}(p) = \mathcal{F}'_2(q_+,q_-,k,|\boldsymbol{p}|,\bar{m}^2_{\pi,k}) - \mathcal{F}'_3(q_+,k,|\boldsymbol{p}|,\bar{m}^2_{\pi,k}),\tag{177}$$

    with

$$q_- = -\sqrt{(p^0 + E(k,\bar{m}^2_{\pi,k}))^2 - \bar{m}^2_{\pi,k}} + |\boldsymbol{p}|,\tag{178}$$

$$q_+ = \sqrt{(p^0 + E(k,\bar{m}^2_{\pi,k}))^2 - \bar{m}^2_{\pi,k}}.\tag{179}$$

III. If $k \le |\boldsymbol{p}|/2$, one has

    (1). when $p^0 > |\boldsymbol{p}|$, then

$$\Im I_{2,k}(p) = 0,\tag{180}$$

    (2). when $p^0 > \left(E(k+|\boldsymbol{p}|,\bar{m}^2_{\pi,k}) - E(k,\bar{m}^2_{\pi,k})\right)$ and $p^0 \le |\boldsymbol{p}|$, then

$$\Im I_{2,k}(p) = \mathcal{F}'_1(q_+,q_-,|\boldsymbol{p}|,\bar{m}^2_{\pi,k}),\tag{181}$$

    with

$$q_- = -\frac{|\boldsymbol{p}|}{2} - \frac{\sqrt{p_0^2(p_0^2-\boldsymbol{p}^2)(p_0^2-\boldsymbol{p}^2-4\bar{m}^2_{\pi,k})}}{2(p_0^2-\boldsymbol{p}^2)},\tag{182}$$

$$q_+ = q_- + |\boldsymbol{p}|;\tag{183}$$

(3). when $p^0 > \left(E(|\boldsymbol{p}|, \bar{m}^2_{\pi,k}) - E(k, \bar{m}^2_{\pi,k})\right)$ and $p^0 \leq \left(E(k+|\boldsymbol{p}|, \bar{m}^2_{\pi,k}) - E(k, \bar{m}^2_{\pi,k})\right)$, then

$$\mathfrak{I}I_{2,k}(p) = \mathcal{F}'_2(q_+, q_-, k, |\boldsymbol{p}|, \bar{m}^2_{\pi,k}) - \mathcal{F}'_3(q_+, k, |\boldsymbol{p}|, \bar{m}^2_{\pi,k}), \qquad (184)$$

with

$$q_- = \sqrt{(p^0 + E(k, \bar{m}^2_{\pi,k}))^2 - \bar{m}^2_{\pi,k}} - |\boldsymbol{p}|, \qquad (185)$$

$$q_+ = q_- + |\boldsymbol{p}|; \qquad (186)$$

(4). when $p^0 > \left(E(k-|\boldsymbol{p}|, \bar{m}^2_{\pi,k}) - E(k, \bar{m}^2_{\pi,k})\right)$ and $p^0 \leq \left(E(|\boldsymbol{p}|, \bar{m}^2_{\pi,k}) - E(k, \bar{m}^2_{\pi,k})\right)$, then

$$\mathfrak{I}I_{2,k}(p) = \mathcal{F}'_2(q_+, q_-, k, |\boldsymbol{p}|, \bar{m}^2_{\pi,k}) - \mathcal{F}'_3(q_+, k, |\boldsymbol{p}|, \bar{m}^2_{\pi,k}), \qquad (187)$$

with

$$q_- = -\sqrt{(p^0 + E(k, \bar{m}^2_{\pi,k}))^2 - \bar{m}^2_{\pi,k}} + |\boldsymbol{p}|, \qquad (188)$$

$$q_+ = \sqrt{(p^0 + E(k, \bar{m}^2_{\pi,k}))^2 - \bar{m}^2_{\pi,k}}; \qquad (189)$$

(5). when $p^0 \leq \left(E(k-|\boldsymbol{p}|, \bar{m}^2_{\pi,k}) - E(k, \bar{m}^2_{\pi,k})\right)$, then

$$\mathfrak{I}I_{2,k}(p) = \mathcal{F}'_1(q_+, q_-, |\boldsymbol{p}|, \bar{m}^2_{\pi,k}), \qquad (190)$$

with

$$q_- = \frac{|\boldsymbol{p}|}{2} + \frac{\sqrt{p_0^2(p_0^2 - \boldsymbol{p}^2)(p_0^2 - \boldsymbol{p}^2 - 4\bar{m}^2_{\pi,k})}}{2(p_0^2 - \boldsymbol{p}^2)}, \qquad (191)$$

$$q_+ = \frac{|\boldsymbol{p}|}{2} - \frac{\sqrt{p_0^2(p_0^2 - \boldsymbol{p}^2)(p_0^2 - \boldsymbol{p}^2 - 4\bar{m}^2_{\pi,k})}}{2(p_0^2 - \boldsymbol{p}^2)}. \qquad (192)$$

Then we consider the case that the curvature of the potential is negative, i.e., $\bar{m}^2_{\pi,k} < 0$, and the functions $\mathfrak{I}I_{1,k}(p)$ and $\mathfrak{I}I_{2,k}(p)$ are modified accordingly. $\mathfrak{I}I_{1,k}(p)$ is given in the following.

I. If $k > |\boldsymbol{p}|$, one has

(1). when $p^0 \geq (E(k+|\boldsymbol{p}|, \bar{m}^2_{\pi,k}) + E(k, \bar{m}^2_{\pi,k}))$, then

$$\mathfrak{I}I_{1,k}(p) = \mathcal{F}_1(q_+, q_-, |\boldsymbol{p}|, \bar{m}^2_{\pi,k}), \qquad (193)$$

with

$$q_- = -\frac{|\boldsymbol{p}|}{2} + \frac{\sqrt{p_0^2(p_0^2 - \boldsymbol{p}^2)(p_0^2 - \boldsymbol{p}^2 - 4\bar{m}^2_{\pi,k})}}{2(p_0^2 - \boldsymbol{p}^2)}, \qquad (194)$$

$$q_+ = q_- + |\boldsymbol{p}|; \qquad (195)$$

(2). when $p^0 < (E(k+|\boldsymbol{p}|, \bar{m}^2_{\pi,k}) + E(k, \bar{m}^2_{\pi,k}))$ and $p^0 > 2E(k, \bar{m}^2_{\pi,k})$, then

$$\mathfrak{I}I_{1,k}(p) = \mathcal{F}_2(q_+, q_-, k, |\boldsymbol{p}|, \bar{m}^2_{\pi,k}) + \mathcal{F}_3(q_+, k, |\boldsymbol{p}|, \bar{m}^2_{\pi,k}), \qquad (196)$$

with

$$q_- = \sqrt{(p^0 - E(k, \bar{m}^2_{\pi,k}))^2 - \bar{m}^2_{\pi,k}} - |\boldsymbol{p}|, \qquad (197)$$

$$q_+ = q_- + |\boldsymbol{p}|; \qquad (198)$$

(3). when $p^0 = 2E(k, \bar{m}_{\pi,k}^2)$, then

$$\Im I_{1,k}(p) = \mathcal{F}_4(p_0, k, |\boldsymbol{p}|, \bar{m}_{\pi,k}^2) ; \tag{199}$$

(4). when $p^0 < 2E(k, \bar{m}_{\pi,k}^2)$, then

$$\Im I_{1,k}(p) = 0 . \tag{200}$$

II. If $|\boldsymbol{p}|/2 < k \le |\boldsymbol{p}|$, one has

(1). when $p^0 \ge (E(k + |\boldsymbol{p}|, \bar{m}_{\pi,k}^2) + E(k, \bar{m}_{\pi,k}^2))$, then

$$\Im I_{1,k}(p) = \mathcal{F}_1(q_+, q_-, |\boldsymbol{p}|, \bar{m}_{\pi,k}^2) , \tag{201}$$

with

$$q_- = -\frac{|\boldsymbol{p}|}{2} + \frac{\sqrt{p_0^2(p_0^2 - \boldsymbol{p}^2)(p_0^2 - \boldsymbol{p}^2 - 4\bar{m}_{\pi,k}^2)}}{2(p_0^2 - \boldsymbol{p}^2)} , \tag{202}$$

$$q_+ = q_- + |\boldsymbol{p}| ; \tag{203}$$

(2). when $p^0 < \left(E(k + |\boldsymbol{p}|, \bar{m}_{\pi,k}^2) + E(k, \bar{m}_{\pi,k}^2)\right)$ and $p^0 \ge \left(E(|\boldsymbol{p}|, \bar{m}_{\pi,k}^2) + E(k, \bar{m}_{\pi,k}^2)\right)$, then

$$\Im I_{1,k}(p) = \mathcal{F}_2(q_+, q_-, k, |\boldsymbol{p}|, \bar{m}_{\pi,k}^2) + \mathcal{F}_3(q_+, k, |\boldsymbol{p}|, \bar{m}_{\pi,k}^2) , \tag{204}$$

with

$$q_- = \sqrt{(p^0 - E(k, \bar{m}_{\pi,k}^2))^2 - \bar{m}_{\pi,k}^2} - |\boldsymbol{p}| , \tag{205}$$

$$q_+ = q_- + |\boldsymbol{p}| ; \tag{206}$$

(3). when $p^0 < \left(E(|\boldsymbol{p}|, \bar{m}_{\pi,k}^2) + E(k, \bar{m}_{\pi,k}^2)\right)$ and $p^0 > 2E(k, \bar{m}_{\pi,k}^2)$, then

$$\Im I_{1,k}(p) = \mathcal{F}_2(q_+, q_-, k, |\boldsymbol{p}|, \bar{m}_{\pi,k}^2) + \mathcal{F}_3(q_+, k, |\boldsymbol{p}|, \bar{m}_{\pi,k}^2) , \tag{207}$$

with

$$q_- = |\boldsymbol{p}| - \sqrt{(p^0 - E(k, \bar{m}_{\pi,k}^2))^2 - \bar{m}_{\pi,k}^2} , \tag{208}$$

$$q_+ = \sqrt{(p^0 - E(k, \bar{m}_{\pi,k}^2))^2 - \bar{m}_{\pi,k}^2} ; \tag{209}$$

(4). when $p^0 = 2E(k, \bar{m}_{\pi,k}^2)$, then

$$\Im I_{1,k}(p) = \mathcal{F}_4(p_0, k, |\boldsymbol{p}|, \bar{m}_{\pi,k}^2) ; \tag{210}$$

(5). when $p^0 < 2E(k, \bar{m}_{\pi,k}^2)$, then

$$\Im I_{1,k}(p) = 0 . \tag{211}$$

III. If $k \le |\boldsymbol{p}|/2$, one has

(1). when $p^0 \geq (E(k+|\boldsymbol{p}|, \bar{m}_{\pi,k}^2) + E(k, \bar{m}_{\pi,k}^2))$, then

$$\Im I_{1,k}(p) = \mathcal{F}_1(q_+, q_-, |\boldsymbol{p}|, \bar{m}_{\pi,k}^2), \tag{212}$$

with

$$q_- = -\frac{|\boldsymbol{p}|}{2} + \frac{\sqrt{p_0^2(p_0^2 - \boldsymbol{p}^2)(p_0^2 - \boldsymbol{p}^2 - 4\bar{m}_{\pi,k}^2)}}{2(p_0^2 - \boldsymbol{p}^2)}, \tag{213}$$

$$q_+ = q_- + |\boldsymbol{p}|; \tag{214}$$

(2). when $p^0 < \left(E(k+|\boldsymbol{p}|, \bar{m}_{\pi,k}^2) + E(k, \bar{m}_{\pi,k}^2)\right)$ and $p^0 \geq \left(E(|\boldsymbol{p}|, \bar{m}_{\pi,k}^2) + E(k, \bar{m}_{\pi,k}^2)\right)$, then

$$\Im I_{1,k}(p) = \mathcal{F}_2(q_+, q_-, k, |\boldsymbol{p}|, \bar{m}_{\pi,k}^2) + \mathcal{F}_3(q_+, k, |\boldsymbol{p}|, \bar{m}_{\pi,k}^2), \tag{215}$$

with

$$q_- = \sqrt{(p^0 - E(k, \bar{m}_{\pi,k}^2))^2 - \bar{m}_{\pi,k}^2} - |\boldsymbol{p}|, \tag{216}$$

$$q_+ = q_- + |\boldsymbol{p}|; \tag{217}$$

(3). when $p^0 < \left(E(|\boldsymbol{p}|, \bar{m}_{\pi,k}^2) + E(k, \bar{m}_{\pi,k}^2)\right)$ and $p^0 \geq 2E(|\boldsymbol{p}|/2, \bar{m}_{\pi,k}^2)$, then

$$\Im I_{1,k}(p) = \mathcal{F}_2(q_+, q_-, k, |\boldsymbol{p}|, \bar{m}_{\pi,k}^2) + \mathcal{F}_3(q_+, k, |\boldsymbol{p}|, \bar{m}_{\pi,k}^2), \tag{218}$$

with

$$q_- = |\boldsymbol{p}| - \sqrt{(p^0 - E(k, \bar{m}_{\pi,k}^2))^2 - \bar{m}_{\pi,k}^2}, \tag{219}$$

$$q_+ = \sqrt{(p^0 - E(k, \bar{m}_{\pi,k}^2))^2 - \bar{m}_{\pi,k}^2}; \tag{220}$$

(4). when $p^0 < 2E(|\boldsymbol{p}|/2, \bar{m}_{\pi,k}^2)$ and $p^0 \geq \left(E(k, \bar{m}_{\pi,k}^2) + E(|\boldsymbol{p}|-k, \bar{m}_{\pi,k}^2)\right)$, then

$$\Im I_{1,k}(p) = \mathcal{F}_2(q_+, q_-, k, |\boldsymbol{p}|, \bar{m}_{\pi,k}^2) + \mathcal{F}_3(q_+, k, |\boldsymbol{p}|, \bar{m}_{\pi,k}^2) - \mathcal{F}_1(q'_+, q'_-, |\boldsymbol{p}|, \bar{m}_{\pi,k}^2), \tag{221}$$

with

$$q_- = |\boldsymbol{p}| - \sqrt{(p^0 - E(k, \bar{m}_{\pi,k}^2))^2 - \bar{m}_{\pi,k}^2}, \tag{222}$$

$$q_+ = \sqrt{(p^0 - E(k, \bar{m}_{\pi,k}^2))^2 - \bar{m}_{\pi,k}^2}, \tag{223}$$

$$q'_- = \frac{|\boldsymbol{p}|}{2} + \frac{\sqrt{p_0^2(p_0^2 - \boldsymbol{p}^2)(p_0^2 - \boldsymbol{p}^2 - 4\bar{m}_{\pi,k}^2)}}{2(p_0^2 - \boldsymbol{p}^2)}, \tag{224}$$

$$q'_+ = \frac{|\boldsymbol{p}|}{2} - \frac{\sqrt{p_0^2(p_0^2 - \boldsymbol{p}^2)(p_0^2 - \boldsymbol{p}^2 - 4\bar{m}_{\pi,k}^2)}}{2(p_0^2 - \boldsymbol{p}^2)}; \tag{225}$$

(5). when $p^0 < \left(E(k, \bar{m}_{\pi,k}^2) + E(|\boldsymbol{p}|-k, \bar{m}_{\pi,k}^2)\right)$, then

$$\Im I_{1,k}(p) = 0. \tag{226}$$

The function $\Im I_{2,k}(p)$ with $\bar{m}^2_{\pi,k} < 0$ is given in the following.

I. If $k > |\boldsymbol{p}|$, one has

(1). when $p^0 > \left( E(k + |\boldsymbol{p}|, \bar{m}^2_{\pi,k}) - E(k, \bar{m}^2_{\pi,k}) \right)$, then

$$\Im I_{2,k}(p) = 0, \tag{227}$$

(2). when $p^0 > |\boldsymbol{p}|$ and $p^0 \leq \left( E(k + |\boldsymbol{p}|, \bar{m}^2_{\pi,k}) - E(k, \bar{m}^2_{\pi,k}) \right)$, then

$$\Im I_{2,k}(p) = \mathcal{F}'_2(q_+, q_-, k, |\boldsymbol{p}|, \bar{m}^2_{\pi,k}) - \mathcal{F}'_3(q_+, k, |\boldsymbol{p}|, \bar{m}^2_{\pi,k}) - \mathcal{F}'_1(q'_+, q'_-, |\boldsymbol{p}|, \bar{m}^2_{\pi,k}), \tag{228}$$

with

$$q_- = \sqrt{(p^0 + E(k, \bar{m}^2_{\pi,k}))^2 - \bar{m}^2_{\pi,k}} - |\boldsymbol{p}|, \tag{229}$$

$$q_+ = q_- + |\boldsymbol{p}|, \tag{230}$$

$$q'_- = -\frac{|\boldsymbol{p}|}{2} + \frac{\sqrt{p_0^2(p_0^2 - \boldsymbol{p}^2)(p_0^2 - \boldsymbol{p}^2 - 4\bar{m}^2_{\pi,k})}}{2(p_0^2 - \boldsymbol{p}^2)}, \tag{231}$$

$$q'_+ = q'_- + |\boldsymbol{p}|; \tag{232}$$

(3). when $p^0 \leq |\boldsymbol{p}|$, then

$$\Im I_{2,k}(p) = \mathcal{F}'_2(q_+, q_-, k, |\boldsymbol{p}|, \bar{m}^2_{\pi,k}) - \mathcal{F}'_3(q_+, k, |\boldsymbol{p}|, \bar{m}^2_{\pi,k}), \tag{233}$$

with

$$q_- = \sqrt{(p^0 + E(k, \bar{m}^2_{\pi,k}))^2 - \bar{m}^2_{\pi,k}} - |\boldsymbol{p}|, \tag{234}$$

$$q_+ = q_- + |\boldsymbol{p}|. \tag{235}$$

II. If $|\boldsymbol{p}|/2 < k \leq |\boldsymbol{p}|$, one has

(1). when $p^0 > \left( E(k + |\boldsymbol{p}|, \bar{m}^2_{\pi,k}) - E(k, \bar{m}^2_{\pi,k}) \right)$, then

$$\Im I_{2,k}(p) = 0, \tag{236}$$

(2). when $p^0 > |\boldsymbol{p}|$ and $p^0 \leq \left( E(k + |\boldsymbol{p}|, \bar{m}^2_{\pi,k}) - E(k, \bar{m}^2_{\pi,k}) \right)$, then

$$\Im I_{2,k}(p) = \mathcal{F}'_2(q_+, q_-, k, |\boldsymbol{p}|, \bar{m}^2_{\pi,k}) - \mathcal{F}'_3(q_+, k, |\boldsymbol{p}|, \bar{m}^2_{\pi,k}) - \mathcal{F}'_1(q'_+, q'_-, |\boldsymbol{p}|, \bar{m}^2_{\pi,k}), \tag{237}$$

with

$$q_- = \sqrt{(p^0 + E(k, \bar{m}^2_{\pi,k}))^2 - \bar{m}^2_{\pi,k}} - |\boldsymbol{p}|, \tag{238}$$

$$q_+ = q_- + |\boldsymbol{p}|, \tag{239}$$

$$q'_- = -\frac{|\boldsymbol{p}|}{2} + \frac{\sqrt{p_0^2(p_0^2 - \boldsymbol{p}^2)(p_0^2 - \boldsymbol{p}^2 - 4\bar{m}^2_{\pi,k})}}{2(p_0^2 - \boldsymbol{p}^2)}, \tag{240}$$

$$q'_+ = q'_- + |\boldsymbol{p}|; \tag{241}$$

(3). when $p^0 > \left(E(|\boldsymbol{p}|, \bar{m}^2_{\pi,k}) - E(k, \bar{m}^2_{\pi,k})\right)$ and $p^0 \le |\boldsymbol{p}|$, then

$$\Im I_{2,k}(p) = \mathcal{F}'_2(q_+, q_-, k, |\boldsymbol{p}|, \bar{m}^2_{\pi,k}) - \mathcal{F}'_3(q_+, k, |\boldsymbol{p}|, \bar{m}^2_{\pi,k}), \tag{242}$$

with

$$q_- = \sqrt{(p^0 + E(k, \bar{m}^2_{\pi,k}))^2 - \bar{m}^2_{\pi,k}} - |\boldsymbol{p}|, \tag{243}$$

$$q_+ = q_- + |\boldsymbol{p}|; \tag{244}$$

(4). when $p^0 \le \left(E(|\boldsymbol{p}|, \bar{m}^2_{\pi,k}) - E(k, \bar{m}^2_{\pi,k})\right)$, then

$$\Im I_{2,k}(p) = \mathcal{F}'_2(q_+, q_-, k, |\boldsymbol{p}|, \bar{m}^2_{\pi,k}) - \mathcal{F}'_3(q_+, k, |\boldsymbol{p}|, \bar{m}^2_{\pi,k}), \tag{245}$$

with

$$q_- = -\sqrt{(p^0 + E(k, \bar{m}^2_{\pi,k}))^2 - \bar{m}^2_{\pi,k}} + |\boldsymbol{p}|, \tag{246}$$

$$q_+ = \sqrt{(p^0 + E(k, \bar{m}^2_{\pi,k}))^2 - \bar{m}^2_{\pi,k}}. \tag{247}$$

III. If $k \le |\boldsymbol{p}|/2$, one has

(1). when $p^0 > \left(E(k + |\boldsymbol{p}|, \bar{m}^2_{\pi,k}) - E(k, \bar{m}^2_{\pi,k})\right)$, then

$$\Im I_{2,k}(p) = 0, \tag{248}$$

(2). when $p^0 > |\boldsymbol{p}|$ and $p^0 \le \left(E(k + |\boldsymbol{p}|, \bar{m}^2_{\pi,k}) - E(k, \bar{m}^2_{\pi,k})\right)$, then

$$\Im I_{2,k}(p) = \mathcal{F}'_2(q_+, q_-, k, |\boldsymbol{p}|, \bar{m}^2_{\pi,k}) - \mathcal{F}'_3(q_+, k, |\boldsymbol{p}|, \bar{m}^2_{\pi,k}) - \mathcal{F}'_1(q'_+, q'_-, |\boldsymbol{p}|, \bar{m}^2_{\pi,k}), \tag{249}$$

with

$$q_- = \sqrt{(p^0 + E(k, \bar{m}^2_{\pi,k}))^2 - \bar{m}^2_{\pi,k}} - |\boldsymbol{p}|, \tag{250}$$

$$q_+ = q_- + |\boldsymbol{p}|, \tag{251}$$

$$q'_- = -\frac{|\boldsymbol{p}|}{2} + \frac{\sqrt{p_0^2(p_0^2 - \boldsymbol{p}^2)(p_0^2 - \boldsymbol{p}^2 - 4\bar{m}^2_{\pi,k})}}{2(p_0^2 - \boldsymbol{p}^2)}, \tag{252}$$

$$q'_+ = q'_- + |\boldsymbol{p}|; \tag{253}$$

(3). when $p^0 > \left(E(|\boldsymbol{p}|, \bar{m}^2_{\pi,k}) - E(k, \bar{m}^2_{\pi,k})\right)$ and $p^0 \le |\boldsymbol{p}|$, then

$$\Im I_{2,k}(p) = \mathcal{F}'_2(q_+, q_-, k, |\boldsymbol{p}|, \bar{m}^2_{\pi,k}) - \mathcal{F}'_3(q_+, k, |\boldsymbol{p}|, \bar{m}^2_{\pi,k}), \tag{254}$$

with

$$q_- = \sqrt{(p^0 + E(k, \bar{m}^2_{\pi,k}))^2 - \bar{m}^2_{\pi,k}} - |\boldsymbol{p}|, \tag{255}$$

$$q_+ = q_- + |\boldsymbol{p}|; \tag{256}$$

(4). when $p^0 > \left(E(k-|\boldsymbol{p}|, \bar{m}^2_{\pi,k}) - E(k, \bar{m}^2_{\pi,k})\right)$ and $p^0 \leq \left(E(|\boldsymbol{p}|, \bar{m}^2_{\pi,k}) - E(k, \bar{m}^2_{\pi,k})\right)$, then

$$\Im I_{2,k}(p) = \mathcal{F}'_2(q_+, q_-, k, |\boldsymbol{p}|, \bar{m}^2_{\pi,k}) - \mathcal{F}'_3(q_+, k, |\boldsymbol{p}|, \bar{m}^2_{\pi,k}), \qquad (257)$$

with

$$q_- = -\sqrt{(p^0 + E(k, \bar{m}^2_{\pi,k}))^2 - \bar{m}^2_{\pi,k}} + |\boldsymbol{p}|, \qquad (258)$$

$$q_+ = \sqrt{(p^0 + E(k, \bar{m}^2_{\pi,k}))^2 - \bar{m}^2_{\pi,k}}; \qquad (259)$$

(5). when $p^0 \leq \left(E(k-|\boldsymbol{p}|, \bar{m}^2_{\pi,k}) - E(k, \bar{m}^2_{\pi,k})\right)$, then

$$\Im I_{2,k}(p) = \mathcal{F}'_1(q_+, q_-, |\boldsymbol{p}|, \bar{m}^2_{\pi,k}), \qquad (260)$$

with

$$q_- = \frac{|\boldsymbol{p}|}{2} + \frac{\sqrt{p_0^2(p_0^2 - \boldsymbol{p}^2)(p_0^2 - \boldsymbol{p}^2 - 4\bar{m}^2_{\pi,k})}}{2(p_0^2 - \boldsymbol{p}^2)}, \qquad (261)$$

$$q_+ = \frac{|\boldsymbol{p}|}{2} - \frac{\sqrt{p_0^2(p_0^2 - \boldsymbol{p}^2)(p_0^2 - \boldsymbol{p}^2 - 4\bar{m}^2_{\pi,k})}}{2(p_0^2 - \boldsymbol{p}^2)}. \qquad (262)$$

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
