# Peer review of "Real-time dynamics of the $O(4)$ scalar theory within the fRG approach"

_SciPost Physics, doi:SciPost Phys. 12, 026 (2022)_

## Round 1 · Referee Report · Anonymous (Referee 3) · 2021-8-19

Report

This work presents first results on the real-time dynamics of the O(4) model in 3+1 dimensions at finite temperature using the FRG approach formulated on the Schwinger-Keldysh contour. In particular, the authors compute the (degenerate) spectral functions in the restored phase, i.e., at temperatures beyond the critical temperature, and the dynamical critical exponent. This work represents an important further development of the real-time FRG approach on the Schwinger-Keldysh contour and a direct extension of recent work in 0+1 dimensions. However, the following remarks and questions need to be addressed before I can recommend publication in SciPost.

Requested changes

1) In Figs. 2, 3 and 4 the labels are very small and barely readable. The authors may want to increase the font size.

2) In Eq. (64) the self-energy is introduced. It may be useful to add an equation that expresses the 2-point function in terms of the self-energy.

3) In section V, the authors state that the flow of $m_\pi^2$ is extracted from (68). It may be useful to add an equation that makes this extraction explicit.

4) The critical temperature in the momentum-dependent calculation (T=20.4 MeV) is considerably smaller than in the momentum-independent case (T=145 MeV). Is there a deeper explanation for this? Is it possible that the “kink” in Fig. 6 is due to some numerical effect?

5) Fig. 5 shows that the squared pion mass changes its sign during the flow in the momentum-dependent case. Does this represent unphysical behavior? A more detailed discussion of this effect would be interesting.

6) The various negative values in the 4-point coupling, the 2-point function, and the spectral function are traced back to the term describing Landau damping. Clearly these negative values are a problem and need to be clearly understood. A more detailed discussion including potential ways to improve the situation would be interesting.

7) Fig. 7 shows values up to k=700 MeV. It may be interesting to add the initial value at the UV cutoff.

8) The left-hand side (lhs) plot in Fig. 12 is a log-log plot while the rhs one is a log-linear plot. It may be better to use the same axes for both plots for better comparability. Also, the rhs plot shows a "0" as well as negative values on the log-axis. This needs to be corrected or discussed since a "0" cannot appear here. Moreover, it would be nice to see the quasi-particle peak also in the rhs plot.

9) Concerning the critical exponent z, I recommend to also cite the recent work by Schweitzer et al. (arXiv: 2007.03374). Also a reference to Schlichting et al. (arXiv: 1908.00912) should be made. In the former work, the dynamic critical behavior of relativistic $Z_2$ theories is studied and classified in terms of the Hohenberg/Halperin scheme. To which of these `model’ classes does the O(4) model studied in the present work belong? Does the extracted value for the dynamical critical exponent agree with this expectation? A more detailed discussion on this point would be useful.

10) The regulator used in the present work, Eq. (20) has no entries in the cc as well as the qq component. Why is the qq-component not regularized? Is the cc-component always zero during the flow, as it should be?

11) Is the location of the threshold of the process phi -> 3 phi consistent with the location of the quasi-particle peak, i.e. is it located at three times the mass (see e.g. Fig. 12)?

  • validity: high
  • significance: top
  • originality: high
  • clarity: high
  • formatting: excellent
  • grammar: good

Author:  Wei-jie Fu  on 2021-10-05  [id 1802]

(in reply to Report 1 on 2021-08-19)

We are grateful to the referee for his comments and suggestions, which we have taken into account in the revised version. In the following, we will respond to the remarks and questions raised by the referee one by one.

1) Thanks for the suggestion. We have enlarged the labels in the relevant figures in the revised version.

2) In the revised version, equation (68) was modified a bit, such that the flow of 2-point function is related to the self-energy.

3) A new equation (74) has been added to make this extraction explicit in the revised version.

4) In order to address this issue more explicitly, we have added a new paragraph at the end of page 8 in the revised version: '' It is interesting to explore underlying reasons accounting for the difference between the momentum independent and dependent results. When the momentum dependence is included, the flow of the effective four-point coupling in Equation (62) is suppressed at finite external momenta. Consequently, the coupling in the flow of the inverse retarded propagator in Equation (70), which contributes mostly around $ |\vec{q}|\sim k$ due to the 3-$d$ momentum integral, is relatively larger than that for the case without momentum dependence. The larger coupling leads to an increased flow of the two-point function as well as a larger meson mass square $m_{\pi,k}^2$ in the infrared, as shown by the blue solid line in the left panel of Figure 5. Hence, lower temperature is required to decrease $m_{\pi,k}^2$ at $k \rightarrow 0$ in order to realize the phase transition. Furthermore, it is found that the kink-like structure of the blue line around about $T=100$ MeV in Figure 6 arises from the fact that when the temperature is below $\sim 100$ MeV, the meson mass square in the region of low $k$ behaves as $m_{\pi,k}^2\sim -k^2$, which results in a small energy factor $E_{\pi,k}(k)$ in Equation (70), cf. also Equation (72), and eventually increases the flow of the inverse retarded propagator even further. That is the reason why the critical temperature in the case with momentum dependence is significantly lower than that without momentum dependence.''

5) No, it doesn't represent unphysical behavior. In the revised version, we have added a discussion in the right column on page 8, trying to make it more clearly: '' Note that the effective potential is broken in the ultraviolet, and the curvature of potential at $\phi_c=0$, i.e., the squared meson mass, cf. Equation (9), is negative when the temperature is below $T_c$. When the temperature is increased above $T_c$, the squared meson mass evolves from the negative to a positive value with the decrease of the RG scale $k$. Therefore, the critical temperature just corresponds to the case that the meson mass square is vanishing at $k \rightarrow 0$.''

6) In the revised version, we have added a paragraph below Fig. 13 to discuss this issue: ''As we have demonstrated above, when the temperature is above and not far away from the critical temperature, the spectral function is positive. However, it is found that when the temperature is quite larger than the critical one, Landau damping contributes a negative value to the spectral function in the regime of small $p_0$. Whether it is an artifact in our computation certainly needs more sophisticated investigations. For instance, the truncation for the flow equation of the four-point vertex as shown in Figure 4 might have to be improved in the region of high temperature, where the momentum dependence should also be encoded for the four-point vertices on the r.h.s. of the flow equation in Figure 4. We hope to report the relevant studies in future work.''

7) The initial value for the imaginary part of the averaged effective vertex in Eq.(75) is vanishing at the UV cutoff. We have added a note below Eq.(75) to address it.

8) We have updated Fig. 12 and used the same axes for both plots. The quasi-particle peak is now more visible in the right plot. Furthermore, in the right panel a symmetric log scale is applied for the $y$-axis in order to take into account both positive and negative values of the spectral function, where the log scale is implicitly translated into a linear one upon crossing the zero point. A sentence has been added in the caption of Fig. 12 to address this issue.

9) Thanks for the suggestions. In the revised version, we have added three references (two mentioned above and also arXiv:hep-ph/9210253). A new paragraph below Eq. (86) was added to discuss this issue: ''Interestingly, this value of the dynamical critical exponent obtained in this work is compatible with a very recent result for Model A in three spatial dimensions, $z=1.92(11)$, obtained from real-time classical-statistical lattice simulations [75]. Here we have used the standard classification for the universality of critical dynamics [14]. However, the critical dynamics of the relativistic $O(4)$ scalar theory should be more closely related to Model G, based on the analysis by Rajagopal and Wilczek [76], see also [77]. The dynamical critical exponent of Model G is $z=3/2$ in three dimensions. Though direct calculation of the dynamical critical exponent for the $O(4)$ model from classical-statistical lattice simulations has not yet arrived at a conclusive result because of errors, it indicates that $z$ is in favor of 2 for small lattice size and $3/2$ for large one [77]. Furthermore, it is also found that the dynamical critical exponent in a relativistic $O(N)$ vector model is close to 2 [63]. In summary, whether the critical dynamics of the relativistic $O(4)$ scalar theory falls into Model A or Model G is still an open question, and more insightful studies are required. ''

10) Vanishing of the cc-component of regulator is due to the fact that there is no such term as $\sim \phi_c^2$ in the effective action in Eq. (1), because the effective action in the Keldysh formalism can be expressed as the difference between those on the forward and backward branches, as shown in Eq. (A1). For the qq-component, regularization is not necessary. The reasons are listed as follows. On the one hand, the Keldysh propagator is related to the retarded and advanced propagators in thermal equilibrium by the fluctuation-dissipation relation as shown in Eq. (36). On the other hand, the dynamics of a system is governed by the retarded and advanced propagators on the level of two-point correlation functions, while the Keldysh propagator represents its statistical properties [18]. It is, therefore, adequate to suppress dynamic quantum fluctuations of different RG scales by encoding regulators only in the retarded and advanced propagators. In the revised version, we have added a paragraph below Eq. (21) to discuss it.

11) Thanks for the question. Following this question, we have inspected the process $\phi \rightarrow 3\phi$ in the spectral function in Fig. 12. This process can be traced back to the imaginary part of the internal momentum averaged effective vertex in Fig. 8, where in the left panel one can see that the sudden rise of the threshold function $I_{1,k}(p-q)$ just corresponds to $p_0=3E_{\pi,k}$. However, its contribution to the spectral function is almost hidden by processes related to, e.g., $I_{1,k}(p+q)$ and $I_{2,k}(p-q)$ as shown in Fig. 9, and hence the process $\phi \rightarrow 3\phi$ is hard to be observed from the spectral function in Fig. 12. In the revised version, we have added several sentences at the end of page 12 to discuss it: ''Furthermore, we have inspected the process $\phi \rightarrow 3\phi$ in the spectral function...''

---

## Round 2 · Referee Report · Anonymous (Referee 1) · 2021-10-18

Report

I thank the authors for taking many of my suggestions into account.

Concerning point 9, the authors now state in the manuscript that

"Though direct calculation of the dynamical critical exponent for the O(4) model from classical-statistical lattice simulations has not yet arrived at a conclusive result because of errors, it indicates that $z$ is in favor of 2 for small lattice size and 3/2 for large ones [77]."

I believe that dropping the last part of this statement reflects the message of [77] better. I therefore recommend to simply write

"Though direct calculation of the dynamical critical exponent for the O(4) model from classical-statistical lattice simulations has not yet arrived at a conclusive result because of errors, it indicates that $z$ is in favor of 2 [77]."

Concerning point 10, I believe that the statement

"For the qq-component, regularization is not necessary"

needs further clarification. In particular, in the paper by Duclut and Delamotte (arXiv: 1611.07301), it is stated that a regularization of the imaginary part of the 2-point function may indeed be necessary. In order to clarify this in the manuscript, I recommend to explicitly state that the regulator is chosen to be real and that therefore the imaginary part of the 2-point function is not regulated. Also a reference to the above mentioned paper should be made in this context.

In the same paper by Duclut and Delamotte, a dynamical critical exponent z close to 2 is found for the O(3) model. The authors may want to point this out when discussing different results on critical exponents and cite the paper also there.
  • validity: -
  • significance: -
  • originality: -
  • clarity: -
  • formatting: -
  • grammar: -

Author:  Wei-jie Fu  on 2021-10-20  [id 1868]

(in reply to Report 1 on 2021-10-18)

We thank the referee for his suggestions in the second report, which we have taken into account in the revised version.

1) Concerning point 9, thanks for the suggestion. We have made relevant modification for the text in the revised version.

2) Concerning point 10, related discussions have been modified below Eq.(21): “Note that in Equation (20) we do not include any regulator for the qq-component, since only the real parts of two-point functions are regulated in this work. This is adequate for cases in thermal equilibrium, where the Keldysh propagator is related to the retarded and advanced propagators by the fluctuation-dissipation relation as shown in Equation (36) in the following. But if the regulator in Equation (17) is extended to the one having a finite imaginary part, as done in some nonequilibrium calculations, e.g. [75], a nonvanishing qq-component of regulators is necessary. ”

Furthermore, we have added the reference [75] (arXiv: 1611.07301), which is also cited in the discussion about the dynamical critical exponent on page 14.

Thanks once more!

---

## Round 2 · Author Response

We are grateful to the referee for his comments and suggestions, which we have taken into account in the revised version. Furthermore, we also respond to the remarks and questions raised by the referee one by one.

---

## Round 2 · List of Changes

1) We have enlarged the labels in Figs. 2, 3, 4.

2) Equation (68) was modified a bit, such that the flow of 2-point function is related to the self-energy.

3) A new equation (74) has been added to make this extraction explicit.

4) We have added a new paragraph at the end of page 8 in the revised version: '' It is interesting to explore underlying reasons accounting for the difference ...''

5) We have added a discussion in the right column on page 8: '' Note that the effective potential is broken in the ultraviolet, and the curvature of potential...''

6) We have added a paragraph below Fig. 13: ''As we have demonstrated above, when the temperature is above and not far away from the critical temperature...''

7) We have added a note below Eq.(75).

8) We have updated Fig. 12 and used the same axes for both plots. The quasi-particle peak is now more visible in the right plot. Furthermore, in the right panel a symmetric log scale is applied for the $y$-axis in order to take into account both positive and negative values of the spectral function, where the log scale is implicitly translated into a linear one upon crossing the zero point. A sentence has been added in the caption of Fig. 12 to address this issue.

9) We have added three references (two mentioned above and also arXiv:hep-ph/9210253). A new paragraph below Eq. (86) was added to discuss this issue: ''Interestingly, this value of the dynamical critical exponent obtained in this work is compatible with a very recent result for Model A in three spatial dimensions... ''

10) We have added a paragraph below Eq. (21) to discuss the regulators.

11) We have added several sentences at the end of page 12: ''Furthermore, we have inspected the process $\phi \rightarrow 3\phi$ in the spectral function...''

---

## Round 3 · Referee Report · Anonymous (Referee 1) · 2021-10-21

Report

I thank the authors for considering my suggestions.

I am not convinced that the formulation concerning "thermal equilibrium" after Eq. (21) is completely correct, but this issue will probably have to be resolved by additional studies in the future. However, it is now clearly formulated what was done in the manuscript.

I therefore recommend publication without further changes.
  • validity: -
  • significance: -
  • originality: -
  • clarity: -
  • formatting: -
  • grammar: -

Author:  Wei-jie Fu  on 2021-10-21  [id 1871]

(in reply to Report 1 on 2021-10-21)

We thank the referee for the recommendation of publication.

We think that the referee still has the question whether it is adequate to regulate only the real part of the 2-point functions in thermal equilibrium. Although this is not quite relevant to the main subject in this manuscript, we would like to address it more clearly.

We think that the answer is "yes". The reason is due to the fact that in thermal equilibrium, the real and imaginary parts are not independent, while they are related to each other through a principal value integral, as shown in e.g., Eq.(B7) and Eq.(B8) in Appendix B. This is also reflected by the fluctuation-dissipation relation.

We hope that this issue is addressed clearly now. Thanks.

---

## Round 3 · Author Response

We thank the referee for his suggestions in the second report, which we have taken into account in the revised version.

---

## Round 3 · List of Changes

1) Minor modification for the text on page 14 is made.

2) We have modified the discussion in the paragraph below Eq.(21), and a new reference (arXiv: 1611.07301) is added.

---

## Editorial Decision

published